# Training Graph Transformers via Curriculum-Enhanced Attention Distillation

**Yisong Huang**[1]**, Jin Li**[1,2]**, Xinlong Chen**[1] **& Yang-Geng Fu**[1]*

[1]College of Computer and Data Science, Fuzhou University, Fuzhou, China
[2]AI Thrust, Information Hub, HKUST (Guangzhou), Guangzhou, China
`huangyisong19@gmail.com jslijin2015@outlook.com`
`fjxinlong@gmail.com fu@fzu.edu.cn`

## Abstract

Recent studies have shown that Graph Transformers (GTs) can be effective for specific graph-level tasks. However, when it comes to node classification, training GTs remains challenging, especially in semi-supervised settings with a severe scarcity of labeled data. Our paper aims to address this research gap by focusing on semi-supervised node classification. To accomplish this, we develop a curriculum-enhanced attention distillation method that involves utilizing a Local GT teacher and a Global GT student. Additionally, we introduce the concepts of in-class and out-of-class and then propose two improvements, *out-of-class entropy* and *top-k pruning*, to facilitate the student's out-of-class exploration under the teacher's in-class guidance. Taking inspiration from human learning, our method involves a curriculum mechanism for distillation that initially provides strict guidance to the student and gradually allows for more out-of-class exploration by a dynamic balance. Extensive experiments show that our method outperforms many state-of-the-art methods on seven public graph benchmarks, proving its effectiveness.

## 1 Introduction

Graph-structured data is ubiquitous, and various real-world systems can be modeled as graphs. In recent years, Graph Neural Networks (GNNs) (Wu et al., 2020; Chen et al., 2020) have shown excellence in addressing graph-related tasks (e.g., node classification (Kipf & Welling, 2017), link prediction (Yun et al., 2021), and graph classification (Errica et al., 2020)) and play a crucial role in applications across diverse domains, including recommender systems (Ying et al., 2018), traffic forecasting (Guo et al., 2019), and molecular property prediction (Kearnes et al., 2016).

GNNs follow the message passing paradigm (Gilmer et al., 2017), which updates the representation of a node by iteratively aggregating messages from neighboring nodes and integrating them with its representation. However, GNNs have some limitations, such as a) difficulty in capturing information from distant nodes effectively and b) high requirements for the reliability of the input graph structure. These limitations restrict, to some extent, the application of GNNs.

Due to the powerful representation capacity of the Transformer architecture (Vaswani et al., 2017), it has found significant success in the fields of natural language processing (NLP) (Kenton & Toutanova, 2019) and computer vision (CV) (Dosovitskiy et al., 2021; Liu et al., 2021). Researchers have been exploring the generalization of Transformers into the field of graphs, commonly referred to as Graph Transformers (GTs). GTs leverage the self-attention mechanism to capture global dependencies among nodes in the graph. This enables each node to adaptively aggregate information not only from its neighbors but also from distant nodes. As a result, this alleviates the limitations of traditional GNNs mentioned above.

GTs can be classified into two main categories: Global Graph Transformers (GGTs) and Local Graph Transformers (LGTs). GGTs, such as Graphormer (Ying et al., 2021) and SAT (Chen et al., 2022), ignore the graph structure and treat each node as a token input to the Transformer module.

---
*Corresponding author.

However, preserving the original graph structure is critical for graph-related tasks, leading to various studies exploring dedicated structural and positional encoding methods. On the other hand, LGTs, such as GT (Dwivedi & Bresson, 2020) and HGT (Hu et al., 2020), perform attention mechanism computations on the original graph structure, with a limited receptive field for each node to capture neighborhood information.

Although GTs have achieved state-of-the-art performance in some graph-level tasks, their effectiveness in node-level tasks, especially semi-supervised node classification, has received limited attention. This work aims to bridge this research gap by enhancing the performance of GTs in semi-supervised node classification tasks. One of the challenges in training GTs is the scarcity of labeled nodes, which can lead to overfitting due to their increased number of learnable parameters. Limited labeled data can make GTs more vulnerable to noise or non-representative nodes, reducing model performance and generalization.

Table 1: Node classification accuracy of GCN, GT, and vanilla Transformer on Cora and Citeseer (mean $\pm$ std %, the best results are bolded).

| Method | Cora | Citeseer |
|---|---|---|
| GCN | **82.06 $\pm$ 0.69** | **71.6 $\pm$ 0.32** |
| GT | 77.04 $\pm$ 0.05 | 66.22 $\pm$ 0.11 |
| naive Transformer | 54.24 $\pm$ 3.16 | 50.60 $\pm$ 3.83 |

To conduct an initial exploratory experiment, we select a set of representative models, including GCN (Kipf & Welling, 2017) (a GNN model), GT (Dwivedi & Bresson, 2020) (an LGT model), and the naive Transformer (Vaswani et al., 2017) (a GGT model), to investigate their node classification accuracy on the Cora and Citeseer datasets. The results are shown in Table 1. It can be observed that GCN outperforms GT and the naive Transformer. There exists a significant performance gap between GT and the naive Transformer. GT can be viewed as a variant of the naive Transformer since it restricts the receptive field of each node to the first-order neighbors, whereas the naive Transformer considers global nodes. Despite the potential of the naive Transformer, its performance falls significantly short. Unleashing the potential of the GTs, particularly the GGTs, is the focus of our research.

To our knowledge, we are the first to propose attention distillation as a solution to address challenges in training GTs, particularly their susceptibility to overfitting. Specifically, we utilize the attention coefficients from each transformer layer in the LGT model (teacher model) to guide the learning of attention coefficients in each transformer layer of the GGT model (student model), enhancing the training process of the GGT model. The student model's learning process consists of two key components: in-class learning, where it benefits from the teacher's guidance, and out-of-class exploration, where it leverages acquired implicit knowledge to seek more beneficial interactions. We propose two improvements, out-of-class entropy and top-k pruning, to encourage the student model to actively explore more refined interactions during out-of-class exploration. However, it's important to note that the distillation process mentioned above has certain limitations. To address them, we introduce "curriculum distillation", an enhanced method incorporating curriculum learning to further unleash the potential of attention distillation.

The contributions of this paper are summarized as follows:

- We propose utilizing attention distillation to enhance the training process of GTs. This involves selecting an LGT as the teacher model and a GGT as the student model for layer-to-layer attention distillation. Furthermore, we propose two significant improvements, out-of-class entropy and top-k pruning, to constrain the student's behavior during out-of-class exploration and enhance its generalization capability.

- To address the limitations of the aforementioned method, we further enhance the proposed distillation method by integrating the concept of curriculum learning, which we refer to as "curriculum distillation".

- We validate the effectiveness of our proposed method on seven graph benchmark datasets. Our method consistently outperforms existing GTs and GNNs, demonstrating enhanced performance and improved generalization capability.

Lastly, we provide an overview of the structure of this paper. We introduce the necessary background and preliminary concepts in Section 2. We present our proposed method in Section 3. Experimental

results and discussions are provided in Section 4. Finally, we conclude the paper in Section 5. For a review of related work on Transformers, knowledge distillation, and curriculum learning in the field of graphs, please refer to Appendix A.

## 2 PRELIMINARIES

### 2.1 PROBLEM DEFINITION

Given a graph $\mathcal{G} = \{\mathcal{V}, \mathcal{E}\}$, where $\mathcal{V}$ and $\mathcal{E}$ denote the sets of nodes and edges, respectively. The graph's structure can be represented by an adjacency matrix $A \in \mathbb{R}^{N \times N}$, where $N$ is the number of nodes. If there exists an edge between node $v_i$ and $v_j$, then $A_{ij} = 1$; otherwise, $A_{ij} = 0$. The nodes are associated with a feature matrix $X \in \mathbb{R}^{N \times d_x}$, where $d_x$ denotes the dimension of the features. Given the labels of a set of nodes $\mathcal{V}_L$, the goal of semi-supervised node classification is to predict the labels of nodes in $\mathcal{V} \backslash \mathcal{V}_L$.

### 2.2 SELF-ATTENTION

Multi-head self-attention (MHA) is the de facto building block of Transformers. For simplicity, we will describe the single-head self-attention module. Given an input sequence $H = [h_1, ..., h_m]^T \in \mathbb{R}^{m \times d_h}$, where $m$ denotes the number of tokens, and $d_h$ denotes the hidden dimension. The self-attention module firstly multiplies the input $H$ by the weight matrices $W^Q \in \mathbb{R}^{d_h \times d_k}$, $W^K \in \mathbb{R}^{d_h \times d_k}$, $W^V \in \mathbb{R}^{d_h \times d_v}$ to obtain queries $Q \in \mathbb{R}^{m \times d_k}$, keys $K \in \mathbb{R}^{m \times d_k}$ and values $V \in \mathbb{R}^{m \times d_v}$:

$$Q = HW^Q, \quad K = HW^K, \quad V = HW^V. \tag{1}$$

The self-attention is then calculated as follows:

$$\text{Attn}(Q, K, V) = \text{Softmax}(\frac{QK^T}{\sqrt{d_k}})V, \tag{2}$$

where $\sqrt{d_k}$ is the scaling factor that avoids gradient vanishing. The attention matrix $\text{Softmax}(\frac{QK^T}{\sqrt{d_k}})$ captures the pairwise interactions among nodes in the graph.

## 3 METHODOLOGY

In this section, we begin by detailing our motivation in Section 3.1. We then provide an in-depth introduction to attention distillation in Section 3.2. Afterward, we present a specific training framework in Section 3.3, which involves selecting an LGT model as the teacher and a GGT model as the student for attention distillation. We also propose two improvements to constrain the student's behavior during out-of-class exploration in Section 3.4. To address the limitations of the above distillation method, we introduce curriculum distillation in Section 3.5.

### 3.1 MOTIVATION

We view the model training process from the perspective of knowledge comprehension. During pre-training, the teacher model learns knowledge from input data and generates downstream predictions based on the acquired knowledge. Considering these predictions as a representation of knowledge, there are inherent limitations even if they can provide guidance to the student model. When facing familiar problems, the student model produces results similar to those of the teacher model. However, when encountering novel or complex problems, the student model often falls short because it merely memorizes the teacher's output without a deeper understanding of the underlying knowledge. Furthermore, the teacher's predictions may contain errors and biases.

Rather than guiding the student to imitate the teacher's predictions, it is more beneficial to help it learn the teacher's "higher-level knowledge understanding" and encourage innovation on this basis. However, extracting the "higher-level knowledge understanding" from the teacher model is a non-trivial task, and considering the model's unique characteristics is essential. The uniqueness of GT models lies in their inherent ability to capture the dependencies among interacting objects (as suggested by (Kim et al., 2022), GTs are good structural learning models). These dependencies support

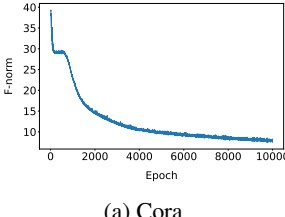
(a) Cora

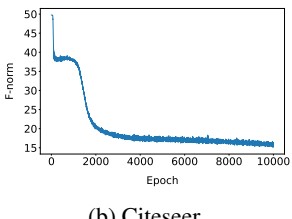
(b) Citeseer

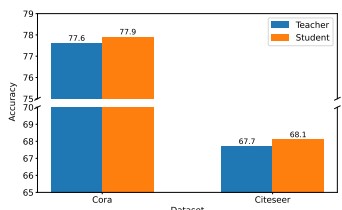

Figure 1: The F-norm dynamics of attention coefficients between GGT student and LGT teacher over time.

Figure 2: Performance comparison: GGT student's compliance to LGT teacher's guidance.

the powerful performance of GT models and establish the foundation for their interpretability. These dependencies are effectively represented through the attention coefficients at each layer of the GT models. We argue that these coefficients accurately represent the teacher's "higher-level knowledge understanding", encompassing its deep comprehension of the problem and underlying reasons for predictions. Transferring this knowledge to students enables them to engage in more fundamental knowledge innovation, resulting in more reliable outcomes.

After discussing the core motivation, we establish the feasibility of our method with three points of additional validation and correction:

a) Feasibility of teacher guidance: This refers to whether the student will follow the teacher's guidance under certain conditions. In experiments, we conduct attention distillation (introduced in Section 3.2), where the student can accurately fit the teacher's attention coefficients (as shown in Figure 1). This indicates a strong ability to retain the teacher's experience and closely adhere to the teacher's guidance.

b) Effectiveness of knowledge comprehension: After learning the knowledge mentioned above, whether the student will make predictions based on the acquired knowledge and achieve comparable performance to the teacher model. Experimental results (as shown in Figure 2) show that the student achieves performance comparable to that of the teacher after following the guidance.

c) Secondary innovativeness: This refers to the student's potential for secondary innovation, surpassing the teacher's performance, which will be thoroughly discussed in Section 4.

## 3.2 ATTENTION DISTILLATION

To enhance clarity, we introduce the teacher and student models as $\mathcal{M}^{[T]}$ and $\mathcal{M}^{[S]}$, respectively. Their attention coefficients for the layers $l \in [1, \ell]$ are denoted as $\alpha^{[T](l)} \in [0, 1]^{N \times N}$ and $\alpha^{[S](l)} \in [0, 1]^{N \times N}$, respectively, where $\ell$ is the number of layers, and $N$ is the number of nodes. These coefficients satisfy $\sum_{j=1}^{N} \alpha_{i,j}^{[T](l)} = \sum_{j=1}^{N} \alpha_{i,j}^{[S](l)} = 1, \forall l \in [1, \ell], i \in [1, N]$. We use the teacher model's attention coefficients to guide the student model's training, which can be formalized as follows:

$$\theta^{*[S]} = \underset{d\left(\alpha^{[S]}, \alpha^{[T]}\right) \leq D_0}{\arg\min} \mathcal{L}^{[S]}(\mathcal{M}^{[S]}(A, X; \theta)_L, Y_L), \tag{3}$$

where $\mathcal{L}^{[S]}(\cdot, \cdot)$ denotes the loss function, such as cross-entropy loss. $\mathcal{M}^{[S]}(A, X; \theta) \in [0, 1]^{N \times C}$ is the probability distribution predicted by the student model, where $C$ is the number of classes. $A \in \mathbb{R}^{N \times N}$ and $X \in \mathbb{R}^{N \times d_x}$ denote the adjacency matrix and node feature matrix of the input graph, respectively, where $d_x$ is the feature dimension. $Y_L$ denotes the ground truth labels for the labeled training nodes in $\mathcal{V}_L$. $\theta^{*[S]}$ denotes the optimal parameters of the student model. $d(\cdot, \cdot)$ denotes a distance metric between two matrices, such as Euclidean distance or Kullback-Leibler divergence. $D_0 \geq 0$ is a predefined hyperparameter that indicates the level of constraint imposed by the teacher model on the student model.

The essence of Equation 3 is introducing the teacher's constraint into the student's optimization process, resulting in a constrained optimization problem. Designing the constraint as a penalty term

(or regularization term) is a common implementation approach:

$$\theta^{*[S]} = \arg\min \mathcal{L}^{[S]}(\mathcal{M}^{[S]}(A, X; \theta)_L, Y_L) + \gamma \cdot P(d(\alpha^{[S]}, \alpha^{[T]}), D_0), \quad (4)$$

where $P(\cdot, \cdot)$ is a penalty function that imposes a penalty when $d(\alpha^{[S]}, \alpha^{[T]})$ exceeds $D_0$, and $\gamma$ is used to control the strength of the penalty. The penalty function $P(\cdot, \cdot)$ can be designed as:

$$P(d(\alpha^{[S]}, \alpha^{[T]}), D_0) = \max(d(\alpha^{[S]}, \alpha^{[T]}) - D_0, 0). \quad (5)$$

Considering that the roles of $\gamma$ and $D_0$ are similar, for simplicity, we set $D_0 = 0$ directly, and the Equation 4 is simplified to:

$$\theta^{*[S]} = \arg\min \mathcal{L}^{[S]}(\mathcal{M}^{[S]}(A, X; \theta)_L, Y_L) + \gamma \cdot d(\alpha^{[S]}, \alpha^{[T]}). \quad (6)$$

### 3.3 DISTILLATION-BASED LGT-GGT TRAINING FRAMEWORK

Our method involves employing a GT teacher with limited representation capacity to guide a GT student with a stronger representation capacity but challenging to optimize. We select a GGT model as the student for its considerable potential, but it tends to overfit without proper guidance. Selecting an LGT model as the teacher stems from two considerations: a) LGT's attention coefficients are more reliable and less prone to overfitting than GGT's; b) Utilizing an LGT teacher to guide a GGT student enhances interpretability, simulating a more effective teacher-student relationship.

Teachers often provide reliable guidance in specific fields in real life, and the "less is more" teaching approach is highly valued. Therefore, we prefer that the teacher model's attention coefficients consider only a subset of nodes rather than all nodes, making LGT a preferable choice. For a specific node $v_i$ in the GGT student, the teacher guides its attention coefficients within its first-order neighborhood (including itself). We define this region as the "in-class" region, denoted as $\mathcal{C}_i^{(\text{in})}$. In real-world datasets, this region is generally relatively small. Node $v_i$ also has a substantial unguided region, defined as the "out-of-class" region, denoted as $\mathcal{C}_i^{(\text{out})} = \mathcal{V} \backslash \mathcal{C}_i^{(\text{in})}$.

For simplicity, we choose a naive LGT model as the teacher and a naive GGT model as the student. Equation 6 can be refined as:

$$\theta^{*[S:GGT]} = \arg\min \mathcal{L}^{[S:GGT]}(\mathcal{M}^{[S:GGT]}(A, X; \theta)_L, Y_L)$$
$$+ \gamma \cdot \frac{1}{\ell} \frac{1}{N} \sum_{l=1}^{\ell} \sum_{i=1}^{N} d(\alpha_{i, \mathcal{C}_i^{(\text{in})}}^{[S:GGT](l)}, \alpha_{i, \mathcal{C}_i^{(\text{in})}}^{[T:LGT](l)}). \quad (7)$$

Furthermore, it's worth noting that our framework can be easily extended to handle multi-head attention.

### 3.4 OUT-OF-CLASS ENTROPY AND TOP-K PRUNING

In this subsection, we propose two improvements to simulate better "teacher-student learning" behavior and improve the student model's performance based on our earlier discussions.

#### 3.4.1 OUT-OF-CLASS ENTROPY

We introduce out-of-class entropy to address the following two issues:

a) Lack of motivation for student innovation: The student model often shows reluctance towards out-of-class exploration. Increasing $\gamma$ in Equation 7 encourages the student to focus on in-class learning, but a small $\gamma$ doesn't motivate sufficient out-of-class exploration. In other words, when $\gamma$ is small, the student still follows the teacher's guidance, as it reduces both supervised and distillation losses. However, while reducing supervised loss, out-of-class exploration may increase distillation loss, resulting in optimization difficulties. This is the fundamental reason for the lack of innovation drive.

b) Distraction of out-of-class attention: During out-of-class exploration, for any given node $v_i \in \mathcal{V}$, it interacts with all nodes in the out-of-class region. This results in a relatively even

distribution of out-of-class attention across multiple nodes, which can lead to potentially risky interactions. To mitigate this, we encourage the student model to focus its out-of-class attention on specific and relevant nodes based on its acquired in-class knowledge.

Specifically, the out-of-class entropy $E_{\text{out}} \in \mathbb{R}$ is designed as follows:

$$
\begin{aligned}
E_{\text{out}} &= \frac{1}{\ell} \frac{1}{N} \sum_{l=1}^{\ell} \sum_{i=1}^{N} \text{Entropy}' \left( \alpha_{i,\mathcal{C}_i^{(\text{out})}}^{[S:GGT](l)} \right) \\
&= -\frac{1}{\ell} \frac{1}{N} \sum_{l=1}^{\ell} \sum_{i=1}^{N} \sum_{j \in \mathcal{C}_i^{(\text{out})}} \alpha_{i,j,\mathcal{C}_i^{(\text{out})}}'^{[S:GGT](l)} log \left( \alpha_{i,j}^{[S:GGT](l)} \right),
\end{aligned}
\tag{8}
$$

where $\alpha_{i,j,\mathcal{C}_i^{(\text{out})}}'^{[S:GGT](l)}$ denotes the normalized out-of-class attention coefficient from node $v_i$ to $v_j$ in the $l$-th layer of the student model, which is defined as:

$$
\alpha_{i,j,\mathcal{C}_i^{(\text{out})}}'^{[S:GGT](l)} = \frac{\alpha_{i,j}^{[S:GGT](l)}}{\sum\limits_{j \in \mathcal{C}_i^{(\text{out})}} \alpha_{i,j}^{[S:GGT](l)} + \varepsilon} \in [0, 1),
\tag{9}
$$

where $\varepsilon$ is a sufficiently small positive constant.

From Equation 8, we can clearly observe that how the out-of-class entropy can alleviate the aforementioned two issues:

a) For any node $v_i \in \mathcal{V}$, let $s_i^{[S:GGT](l)} = \sum\limits_{j \in \mathcal{C}_i^{(\text{out})}} \alpha_{i,j}^{[S:GGT](l)} + \varepsilon \in (0, 1)$, which represents the total attention allocated by node $v_i$ in the out-of-class exploration. If this value is smaller, $E_{\text{out}}$ tends to be larger. Therefore, minimizing $E_{\text{out}}$ will drive the student to engage in out-of-class exploration.

b) If $s_i^{[S:GGT](l)}$ is kept constant, minimizing $E_{\text{out}}$ results in the sparsity of out-of-class attention coefficient distribution $\{\alpha_{i,j}^{[S:GGT](l)}\}_{j \in \mathcal{C}_i^{(\text{out})}}$. The detailed proof is provided in Appendix B.

We incorporate the out-of-class entropy into the loss function, and Equation 7 is improved as:

$$
\begin{aligned}
\theta^{*[S:GGT]} &= \arg\min \mathcal{L}^{[S:GGT]}(\mathcal{M}^{[S:GGT]}(A, X; \theta)_L, Y_L) \\
&+ \gamma \cdot \frac{1}{\ell} \frac{1}{N} \sum_{l=1}^{\ell} \sum_{i=1}^{N} d(\alpha_{i,\mathcal{C}_i^{(in)}}^{[S:GGT](l)}, \alpha_{i,\mathcal{C}_i^{(in)}}^{[T:LGT](l)}) + \beta \cdot E_{\text{out}},
\end{aligned}
\tag{10}
$$

where $\beta$ is a hyperparameter that controls the degree of out-of-class drive.

### 3.4.2 Top-K Pruning

For each node $v_i \in \mathcal{V}$, its out-of-class region $\mathcal{C}_i^{(\text{out})}$ is generally large, containing numerous nodes due to the low average degree in real graph datasets. Using GT models for node classification tasks can risk overfitting when normalizing attention coefficients, as seen in (Wu et al., 2022), due to the large number of nodes involved in the normalization process.

To address the aforementioned issue, we propose top-k pruning, a simple yet practical improvement. It prunes the out-of-class attention coefficients before computing out-of-class entropy. Specifically, we first obtain $\{\alpha_{i,j}^{[S:GGT](l)}\}_{j \in \mathcal{C}^{(\text{out})}}$, and then retain only the top $k$ largest attention coefficients while setting the rest to 0,

$$
\begin{aligned}
\alpha_{i,j}^{[S:GGT](l)} &= \begin{cases} \alpha_{i,j}^{[S:GGT](l)}, & j \in \text{Index}^{\{k\}}, \\ 0, & j \notin \text{Index}^{\{k\}}, \end{cases} \quad \forall l \in [1, \ell], i \in [1, N], \\
\text{Index}^{\{k\}} &= \arg\text{top}_k \left( \{\alpha_{i,j}^{[S:GGT](l)}\}_{j \in \mathcal{C}_i^{(\text{out})}} \right),
\end{aligned}
\tag{11}
$$

where the $\arg \mathrm{top}_k \left( \cdot \right)$ operation selects the indices corresponding to the top $k$ largest attention coefficients in the out-of-class region for each node. $k$ is a hyperparameter used to control the number of nodes retained in the out-of-class region.

## 3.5 CURRICULUM DISTILLATION

Reviewing the previous method, the student model maintains a constant strategy, keeping $\gamma$ and $\beta$ fixed throughout the learning process. We aim for larger values of $\gamma$ and $\beta$ to ensure that the student receives effective guidance from the teacher while maintaining sufficient motivation for out-of-class exploration. However, there exists a competition in the allocation of attention between in-class learning and out-of-class exploration. When both $\gamma$ and $\beta$ are large, conflicts arise, leading to the following two issues:

- Inadequate out-of-class exploration due to insufficient attention allocation.
- Excessive out-of-class exploration without a proper understanding of in-class knowledge, hindering both in-class learning and out-of-class exploration.

Therefore, attention allocation should evolve dynamically during training to enhance the student model's performance. Initially, priority should be given to in-class knowledge, and as it consolidates, attention to out-of-class exploration should gradually increase, ultimately achieving a balanced state.

Based on the above motivation, we propose "curriculum distillation", which integrates the "curriculum" concept into the distillation process. It allows for a structured teaching approach where the teacher initially provides strict supervision and guidance to the student, then gradually allows the student to engage in out-of-class exploration and innovation freely.

We present an abstract framework that dynamically manages the hyperparameter $\gamma$:

$$\gamma^{(t)} = \mathrm{Schedule}_\gamma(t), \quad \forall t \in [1, T_{\mathrm{epoch}}], \tag{12}$$

where $T_{\mathrm{epoch}}$ denotes the total number of epochs, $\gamma^{(t)}$ denotes the value of $\gamma$ at the $t$-th epoch, and $\mathrm{Schedule}_\gamma(\cdot)$ is a non-strictly decreasing schedule function.

We also need to manage the hyperparameter $\beta$ dynamically for out-of-class exploration by designing a similar schedule function, except that this function is non-strictly increasing:

$$\beta^{(t)} = \mathrm{Schedule}_\beta(t), \quad \forall t \in [1, T_{\mathrm{epoch}}]. \tag{13}$$

Equation 12 and Equation 13 can be implemented differently. Specifically, we adopt the following design to calculate $\gamma^{(t)}$:

$$
\begin{aligned}
\gamma^{(t)} &= \mathrm{Schedule}_\gamma \left( t, \mathrm{ep}^{(\mathrm{begin})}, \mathrm{ep}^{(\mathrm{end})}, \gamma^{(\mathrm{begin})}, \gamma^{(\mathrm{end})} \right), \quad \forall t \in [1, T_{\mathrm{epoch}}] \\
&= \begin{cases}
\gamma^{(\mathrm{begin})}, & 1 \le t < \mathrm{ep}^{(\mathrm{begin})} \\
\gamma^{(\mathrm{begin})} + \left( 1 + \cos \left( \pi \cdot \left( 1 - \frac{t - \mathrm{ep}^{(\mathrm{begin})}}{\mathrm{ep}^{(\mathrm{end})} - \mathrm{ep}^{(\mathrm{begin})}} \right) \right) \right) \cdot \frac{\gamma^{(\mathrm{end})} - \gamma^{(\mathrm{begin})}}{2}, & \mathrm{ep}^{(\mathrm{begin})} \le t \le \mathrm{ep}^{(\mathrm{end})} \\
\gamma^{(\mathrm{end})}, & \mathrm{ep}^{(\mathrm{end})} < t \le T_{\mathrm{epoch}}
\end{cases}
\end{aligned} \tag{14}
$$

Within the epoch interval $[\mathrm{ep}^{(\mathrm{begin})}, \mathrm{ep}^{(\mathrm{end})}]$, $\gamma$ varies, while its values remain unchanged outside this interval. The hyperparameters $\gamma^{(\mathrm{begin})}$ and $\gamma^{(\mathrm{end})}$ represent the initial and final values of $\gamma$, respectively, with $\gamma^{(\mathrm{begin})} > \gamma^{(\mathrm{end})}$ indicating non-strictly decreasing. Similarly, $\beta$ follows Equation 14, sharing the same epoch interval $[\mathrm{ep}^{(\mathrm{begin})}, \mathrm{ep}^{(\mathrm{end})}]$ with $\gamma$ but having different initial and final values, denoted as $\beta^{(\mathrm{begin})}$ and $\beta^{(\mathrm{end})}$, respectively, where $\beta^{(\mathrm{begin})} < \beta^{(\mathrm{end})}$.

For an intuitive understanding, we present the curves illustrating the variations of hyperparameters $\gamma$ and $\beta$ across epochs under different curriculum designs in Appendix C.

## 4 EXPERIMENTS

### 4.1 EXPERIMENTAL SETUP

We briefly introduce the datasets and baselines in experiments. More details are in Appendix E.

**Datasets**. We evaluate our method on seven benchmark datasets, including citation network datasets Cora, Citeseer, and Pubmed (Sen et al., 2008); Actor co-occurrence network dataset (Chien et al., 2021); and WebKB datasets (Pei et al., 2020) including Cornell, Texas, and Wisconsin. We apply the standard splits for the citation network datasets, as in the previous work (Kipf & Welling, 2017). For the remaining datasets, we set the train-validation-test split as 48%/32%/20%.

**Baselines**. We compare our method against ten baseline methods, including five GNN methods, i.e., GCN (Kipf & Welling, 2017), GAT (Veličković et al., 2018), GraphSAGE (Hamilton et al., 2017), SuperGAT$_{SD}$ (Kim & Oh, 2021), and APPNP (Gasteiger et al., 2019), along with five Graph Transformers, i.e., GT (Dwivedi & Bresson, 2020), SAN (Kreuzer et al., 2021), Graphormer (Ying et al., 2021), ANS-GT (Zhang et al., 2022) and NodeFormer (Wu et al., 2022).

## 4.2 Node Classification Performance

We conduct five random runs and report the average and standard deviation of the experimental results. The results are shown in Table 2. We have the following observations: a) The proposed method outperforms all models and shows significant improvements on multiple datasets compared to the baselines. For example, it achieves a relative improvement of 6.70% over GCN on Citeseer. Compared to Graphormer, it achieves relative improvements of 34.27% on Cora and 22.85% on Cornell. b) GTs perform less effectively on node classification tasks than GNNs, which can be attributed to two reasons: First, GTs have more parameters than GNNs, making them more challenging to train with limited labeled data. Second, GTs lack local inductive biases, limiting their ability to leverage the inherent structural information in the graph effectively.

Table 2: Node classification performance (mean $\pm$ std%, the best results are bolded and the runner-ups are underlined).

| Method | Cora | Citeseer | Pubmed | Actor | Cornell | Texas | Wisconsin |
|---|---|---|---|---|---|---|---|
| GCN | $82.06 \pm 0.69$ | $71.60 \pm 0.32$ | $79.58 \pm 0.26$ | $27.88 \pm 0.07$ | $53.51 \pm 1.21$ | $69.19 \pm 1.48$ | $57.25 \pm 1.25$ |
| GAT | $82.84 \pm 0.67$ | $72.28 \pm 0.72$ | $78.52 \pm 0.32$ | $28.71 \pm 0.25$ | $55.14 \pm 0.48$ | $68.65 \pm 1.48$ | $58.82 \pm 2.40$ |
| GraphSAGE | $81.40 \pm 0.52$ | $71.68 \pm 0.13$ | $78.50 \pm 0.44$ | $36.24 \pm 0.39$ | $63.78 \pm 2.42$ | $75.14 \pm 1.21$ | $76.08 \pm 0.88$ |
| SuperGAT$_{SD}$ | $82.70 \pm 0.60$ | $\underline{72.50 \pm 0.80}$ | $\underline{81.30 \pm 0.50}$ | $30.18 \pm 0.25$ | $54.59 \pm 1.21$ | $69.73 \pm 1.21$ | $58.04 \pm 1.07$ |
| APPNP | $\underline{84.10 \pm 0.43}$ | $72.14 \pm 0.22$ | $80.02 \pm 0.19$ | $33.47 \pm 0.38$ | $61.08 \pm 1.48$ | $71.35 \pm 1.48$ | $65.10 \pm 0.88$ |
| GT | $77.58 \pm 0.22$ | $66.96 \pm 0.36$ | $76.48 \pm 0.13$ | $37.15 \pm 0.40$ | $61.62 \pm 2.26$ | $74.60 \pm 1.48$ | $71.76 \pm 1.75$ |
| SAN | $77.60 \pm 0.23$ | $68.64 \pm 0.92$ | $76.62 \pm 0.22$ | $37.79 \pm 0.20$ | $63.24 \pm 1.48$ | $75.14 \pm 2.26$ | $\underline{77.25 \pm 1.75}$ |
| Graphormer | $63.08 \pm 0.27$ | $61.08 \pm 0.04$ | OOM | OOM | $62.70 \pm 2.26$ | $76.76 \pm 1.48$ | $72.16 \pm 0.88$ |
| ANS-GT | $77.68 \pm 0.81$ | $64.16 \pm 1.16$ | $77.98 \pm 1.38$ | $\underline{38.29 \pm 0.61}$ | $74.92 \pm 1.86$ | $76.22 \pm 2.26$ | $76.47 \pm 1.96$ |
| Nodeformer | $79.02 \pm 0.57$ | $69.66 \pm 0.13$ | $76.06 \pm 0.98$ | $34.80 \pm 0.48$ | $68.11 \pm 1.21$ | $\underline{77.84 \pm 1.71}$ | $76.47 \pm 4.80$ |
| Ours | $\mathbf{84.70 \pm 0.56}$ | $\mathbf{76.40 \pm 0.70}$ | $\mathbf{83.10 \pm 0.42}$ | $\mathbf{39.40 \pm 0.09}$ | $\mathbf{77.03 \pm 1.90}$ | $\mathbf{85.58 \pm 1.56}$ | $\mathbf{81.95 \pm 0.87}$ |

Table 3: Distillation performance comparison.

| Model | Cora | Citeseer | Pubmed | Actor | Cornell | Texas | Wisconsin |
|---|---|---|---|---|---|---|---|
| Teacher (LGT) | $\underline{77.40 \pm 0.56}$ | $\underline{68.05 \pm 1.06}$ | $\underline{79.05 \pm 0.63}$ | $\underline{36.87 \pm 0.14}$ | $\underline{68.46 \pm 1.56}$ | $76.57 \pm 1.56$ | $75.48 \pm 1.38$ |
| Student (GGT) | $54.24 \pm 3.16$ | $50.60 \pm 3.83$ | $72.50 \pm 0.87$ | $35.97 \pm 0.29$ | $65.76 \pm 5.62$ | $\underline{77.03 \pm 2.70}$ | $\underline{77.55 \pm 3.93}$ |
| Ours | $\mathbf{84.70 \pm 0.56}$ | $\mathbf{76.40 \pm 0.70}$ | $\mathbf{83.10 \pm 0.42}$ | $\mathbf{39.40 \pm 0.09}$ | $\mathbf{77.03 \pm 1.90}$ | $\mathbf{85.58 \pm 1.56}$ | $\mathbf{81.95 \pm 0.87}$ |

## 4.3 Distillation Comparative Studies

Our method is meaningful only if the student model's performance improves with teacher guidance and surpasses the teacher model's after training. We summarize the performance comparison of our proposed method (Ours), the teacher model (LGT), and the unguided student model (GGT) on all seven datasets, including average accuracy and standard deviation, as shown in Table 3.

From Table 3, we observe the following: a) On average, our proposed method exhibits an absolute improvement of 6.61% compared to the teacher model. Specifically, on the Cornell dataset, a relative performance of 12.51% is achieved; b) On average, our method achieves an absolute improvement of 13.50% compared to the unguided student model. Specifically, a relative performance improvement of 56.16% is observed on the Cora dataset. Therefore, the proposed method significantly aids the training process of the GGT student model despite both the selected teacher and student models being naive GT models with notable limitations.

## 4.4 ABLATION STUDIES

We conduct ablation studies to verify and understand the roles of different internal modules in our method. The results are shown in Table 4.

**Module Ablation**. "w.o. $\gamma$" and "w.o. $\beta$" indicate the removal of the distillation loss and the out-of-class entropy by setting them to 0, respectively. "w.o. top-k" indicates the removal of the top-k pruning. Our proposed method exhibits an average absolute performance of 34.29%, 9.05%, and 6.29% across these three variants on all datasets, demonstrating the significance and design rationality of each module.

**Curriculum Ablation**. "w.o. curriculum" indicates the removal of the curriculum learning framework. "w.o. epoch design" indicates the removal of the fine-grained interval range of the curriculum by setting $ep^{begin} = 0$ and $ep^{end} = 500$. Compared to these two variants, our proposed method achieves an average absolute performance improvement of 10.01% and 1.74%, respectively, indicating the importance and design rationality of the curriculum framework.

**Distillation Ablation**. "GGT $\rightarrow$ GGT" indicates using a GGT model as the teacher instead of an LGT to guide the GGT student. "Local neighborhood uniform distribution" indicates degrading the teacher's attention coefficients at each layer to a fixed first-order neighborhood uniform distribution. "Global uniform distribution" indicates degrading the teacher's attention coefficients at each layer to a fixed global uniform distribution. Compared to these three variants, our proposed method achieves average absolute performance improvements of 12.88%, 15.13%, and 32.96%, respectively, on all datasets, demonstrating the rationality of the selected teacher model and the significant importance of the attention distillation compared to the degraded regularization.

Table 4: The performance evaluation of variants of our method.

| Categories | Method | Cora | Citeseer | Pubmed | Actor | Cornell | Texas | Wisconsin |
|---|---|---|---|---|---|---|---|---|
| Module Ablation | w.o. $\gamma$ | 31.90 | 18.10 | 40.70 | 25.59 | 54.05 | 64.86 | 52.94 |
| | w.o. $\beta$ | 81.20 | 72.70 | 81.00 | 36.18 | 59.46 | 67.57 | 66.67 |
| | w.o. top-k | 82.00 | 71.20 | 82.85 | 36.18 | 67.57 | 75.68 | 68.62 |
| Curriculum Ablation | w.o. curriculum | 82.10 | 72.10 | 79.90 | 34.74 | 64.86 | 67.57 | 56.80 |
| | w.o. epoch design | 83.90 | 74.90 | 81.30 | 38.75 | 72.97 | 83.78 | 80.39 |
| Distillation Ablation | GGT $\rightarrow$ GGT | 56.60 | 56.23 | OOM | OOM | 69.37 | 77.70 | 81.37 |
| | Local neighborhood uniform distribution | 81.70 | 72.50 | 79.70 | 26.11 | 51.35 | 54.05 | 56.86 |
| | Global uniform distribution | 31.90 | 18.10 | OOM | 25.46 | 54.05 | 64.86 | 52.94 |
| | Ours | **84.70** | **76.40** | **83.10** | **39.40** | **77.03** | **85.58** | **81.95** |

## 4.5 ADDITIONAL ANALYSIS

We conduct a comparison between our method and other traditional knowledge distillation methods, along with hyperparameter studies, interpretability studies, and visualization. For detailed experimental results and analysis, please refer to Appendix F.

Additionally, we demonstrate the extension of our method to GNNs, such as GCN, by performing a distillation experiment from GCN to GGT. For a comprehensive examination of results and analysis, please refer to Appendix H.

## 5 CONCLUSION

In this paper, we propose a novel curriculum-enhanced attention distillation method to improve the training process of Graph Transformers, which is both simpler and more interpretable. By utilizing the attention coefficients within the Transformer layers as knowledge representations, we achieve a more efficient knowledge transfer from the LGT teacher to the GGT student. Furthermore, we introduce out-of-class entropy and top-k pruning as enhancements to promote active out-of-class exploration by the student under the teacher's guidance, enabling the student to surpass the teacher's performance and unleash its potential. Moreover, inspired by curriculum learning, we introduce curriculum distillation to capture the dynamic evolution of the attention distribution between in-class learning and out-of-class exploration during the student's learning process, further improving performance. Experimental results on multiple datasets demonstrate the effectiveness of our method in improving the training process of GTs and enhancing their generalization capability.

## 6 ACKNOWLEDGEMENTS

This research was supported by the University-Industry Cooperation Project of Fujian Province, China (2023H6008) and the National Natural Science Foundation of China (12271098).

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

# A   RELATED WORK

## A.1   TRANSFORMERS FOR GRAPH

The Transformer architecture (Vaswani et al., 2017) has achieved great success in many fields, such as NLP and CV, stimulating researchers' interest in generalizing it to graph-structure data. GTs are designed to solve graph-related problems such as node and graph classification. Some pioneering studies on GTs focus on node classification and have achieved competitive performance compared to GNNs. ANS-GT (Zhang et al., 2022) adaptively selects a node sampling strategy from four strategies (1-hop neighbors, 2-hop neighbors, KNN, and PPR) through a multi-armed bandit mechanism to select the most informative nodes as input to the Transformer module. Graph-Bert (Zhang et al., 2020) limits the receptive field of each target node to the top-k nodes in the intimacy matrix calculated based on the PageRank algorithm. NodeFormer (Wu et al., 2022) approximates pairwise similarity through kernel functions, and it also uses Gumble-Softmax to enable differentiable optimization of graph structures. Compared to the above methods, our method is more concise and interpretable.

## A.2   KNOWLEDGE DISTILLATION

Knowledge distillation was first proposed by Hinton (Hinton et al., 2015). It is a "teacher-student" network training method that transfers the knowledge learned by a large model with a complex architecture and sufficient capacity (called the teacher model) to a small model with a simple architecture and insufficient capacity (called the student model) to improve its performance.

Traditional knowledge distillation typically involves two main stages. Initially, the teacher model undergoes pre-training, and its output or intermediate layer representations guide the student model. For instance, KD (Hinton et al., 2015) utilizes the teacher's logits as soft supervision, while Fit-Nets (Romero et al., 2014) leverages the teacher model's intermediate layer features as hints for knowledge transfer, and AT (Zagoruyko & Komodakis, 2016) employs spatial attention maps for guidance. Some current research focuses on enhancing the effectiveness of knowledge distillation. NORM (Liu et al., 2023) focuses on many-to-one representation matching, introducing a Feature Transform module for this purpose. DisWOT (Dong et al., 2023) addresses model capacity gaps by using a neural network architecture without training to find the optimal student architecture. To bridge the performance gap between offline and online distillation methods, SHAKE (Li & Jin, 2022) combines both approaches using a proxy teacher model with a shadow head and a student model for bidirectional distillation. Furthermore, there are variants of knowledge distillation frameworks. For example, Tf-FD (Li, 2022) introduces a teacher-free feature distillation method that utilizes salient features within the student model for distillation.

Beyond its widespread use in the fields of CV and NLP, knowledge distillation is also gradually being introduced into graph-related tasks. LSP (Yang et al., 2020) is the first knowledge distillation framework tailored for homogeneous GNNs. It develops a local structure preserving module (LSP) in GNNs to capture structural information and transfer the structural knowledge from the teacher to the student by minimizing the distance between their distributions. HIRE (Liu et al., 2022) aims to preserve relational knowledge in heterogeneous GNNs, which distills the correlation knowledge between a single node soft label and different node types into the student model. G-CRD (Joshi et al., 2022) employs a contrastive learning method to implicitly preserve the global topology by aligning student node embeddings with teacher node embeddings in a shared representation space.

Differing from existing knowledge distillation methods that aim at model compression, our method is designed to address the challenging task of training Graph Transformers in semi-supervised scenarios. The representation capacity of the teacher model we employ is weaker than that of the student model. In this situation, the teacher's guidance can be seen as a constraint on the student. However, the student model possesses a significant potential that requires proper guidance for its full realization. We intend to utilize knowledge distillation to alleviate issues encountered by the student model during training, such as overfitting, and to improve the quality of learned representations.

## A.3 Curriculum Learning

As a mature strategy for training machine learning models, curriculum learning (Wang et al., 2021a; Soviany et al., 2022), has gained widespread adoption in recent years across various fields, including NLP(Sachan & Xing, 2016; Xu et al., 2020) and CV (Jiang et al., 2018; Zhou et al., 2020). It involves organizing samples from easy to difficult (Bengio et al., 2009), enabling the model to gradually learn from simple to complex knowledge, similar to human learning, and enhancing the learning effect. Curriculum learning for graph-structure data is an emerging technology that aims to improve the performance of graph-related tasks. To solve graph classification tasks, CurGraph (Wang et al., 2021b) uses neural density estimation to model the distribution of graph-level embeddings. Then, it calculates the difficulty scores of graphs based on their embeddings' intra-class and inter-class distributions. CuCo (Chu et al., 2021) combines curriculum learning with graph contrastive learning. It uses a similarity measure function to score the negative samples and then ranks the negative samples from easy to hard according to the scores. CuCo starts with easier-to-distinguish negative samples during training and progressively introduces more challenging ones.

## B Proof: Minimizing Out-of-Class Entropy for Sparse Attention Coefficients

In the main text, we introduce the out-of-class entropy. Notably, the normalized out-of-class attention coefficients $\alpha_{i,j,\mathcal{C}_i^{(\text{out})}}^{\prime[S:GGT](l)}$ approximately satisfies:

$$\forall l \in [1, \ell], i \in [1, N], \sum_{j \in \mathcal{C}_i^{(\text{out})}} \alpha_{i,j,\mathcal{C}_i^{(\text{out})}}^{\prime[S:GGT](l)} = 1. \tag{15}$$

Here, we prove that when the sum of out-of-class attention coefficients for node $v_i$ is held constant, i.e., $s_i^{[S:GGT](l)} = \sum_{j \in \mathcal{C}_i^{(\text{out})}} \alpha_{i,j}^{[S:GGT](l)} + \varepsilon = s_0 \in (0, 1)$, minimizing the out-of-class entropy $E_{\text{out}}$ leads to a sparser distribution of out-of-class attention coefficients $\{\alpha_{i,j}^{[S:GGT](l)}\}_{j \in \mathcal{C}_i^{(\text{out})}}$.

**Proof**.

$$
\begin{aligned}
\text{Entropy}' \left( \alpha_{i,\mathcal{C}_i^{(\text{out})}}^{[S:GGT](l)} \right) &= \sum_{j \in \mathcal{C}_i^{(\text{out})}} \alpha_{i,j,\mathcal{C}_i^{(\text{out})}}^{\prime[S:GGT](l)} \cdot \log \left( \alpha_{i,j}^{[S:GGT](l)} \right) \\
&= \sum_{j \in \mathcal{C}_i^{(\text{out})}} \alpha_{i,j,\mathcal{C}_i^{(\text{out})}}^{\prime[S:GGT](l)} \cdot \log \left( \frac{\alpha_{i,j}^{[S:GGT](l)}}{s_i^{[S:GGT](l)}} \cdot s_i^{[S:GGT](l)} \right) \\
&= \sum_{j \in \mathcal{C}_i^{(\text{out})}} \alpha_{i,j,\mathcal{C}_i^{(\text{out})}}^{\prime[S:GGT](l)} \cdot \left( \log \left( \frac{\alpha_{i,j}^{[S:GGT](l)}}{s_i^{[S:GGT](l)}} \right) + \log \left( s_i^{[S:GGT](l)} \right) \right) \\
&= \sum_{j \in \mathcal{C}_i^{(\text{out})}} \alpha_{i,j,\mathcal{C}_i^{(\text{out})}}^{\prime[S:GGT](l)} \cdot \log \left( \frac{\alpha_{i,j}^{[S:GGT](l)}}{s_i^{[S:GGT](l)}} \right) + \sum_{j \in \mathcal{C}_i^{(\text{out})}} \alpha_{i,j,\mathcal{C}_i^{(\text{out})}}^{\prime[S:GGT](l)} \cdot \log \left( s_i^{[S:GGT](l)} \right) \\
&= \sum_{j \in \mathcal{C}_i^{(\text{out})}} \alpha_{i,j,\mathcal{C}_i^{(\text{out})}}^{\prime[S:GGT](l)} \cdot \log \left( \alpha_{i,j,\mathcal{C}_i^{(\text{out})}}^{\prime[S:GGT](l)} \right) + \log \left( s_i^{[S:GGT](l)} \right) \cdot \sum_{j \in \mathcal{C}_i^{(\text{out})}} \alpha_{i,j,\mathcal{C}_i^{(\text{out})}}^{\prime[S:GGT](l)} \\
&\approx \sum_{j \in \mathcal{C}_i^{(\text{out})}} \alpha_{i,j,\mathcal{C}_i^{(\text{out})}}^{\prime[S:GGT](l)} \cdot \log \left( \alpha_{i,j,\mathcal{C}_i^{(\text{out})}}^{\prime[S:GGT](l)} \right) + \log \left( s_i^{[S:GGT](l)} \right) \cdot 1 \\
&\approx \text{Entropy} \left( \{\alpha_{i,j,\mathcal{C}_i^{(\text{out})}}^{\prime[S:GGT](l)}\}_{j \in \mathcal{C}_i^{(\text{out})}} \right) + \log(s_0) \\
&\propto \text{Entropy}(\alpha_{i,\mathcal{C}_i^{(\text{out})}}^{\prime[S:GGT](l)}).
\end{aligned}
\tag{16}
$$

Thus, based on the principle that minimizing entropy corresponds to a sparser probability distribution, we conclude that this result is valid, thereby completing the proof.

## C    Variations of Hyperparameters under Curriculum Design

Here, we present the curves of hyperparameters $\gamma$ and $\beta$ as they evolve over epochs (under different curriculum designs), as shown in Figure 3.

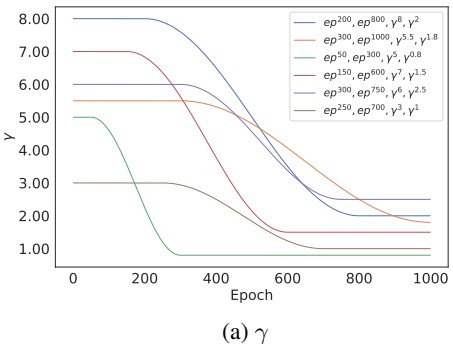

(a) $\gamma$

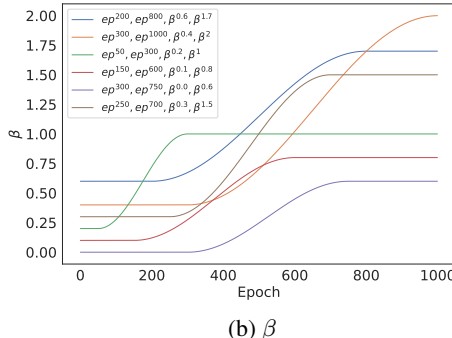

(b) $\beta$

Figure 3: The variation curves of hyperparameters $\gamma$ and $\beta$ under different curriculum designs.

## D    Complexity Analysis

The time and space complexity of pre-training the teacher model can be neglected compared to training the student model, and curriculum learning does not inherently increase complexity. Therefore, we only discuss the time and space complexity of the student model.

The time complexity of a single forward propagation of the attention module is $O(nd^2 + n^2d) = O(n^2d)$, and the space complexity is $O(nd+n^2)$. The feed-forward network (FFN) requires $O(nd^2)$ time and $O(nd)$ space per layer. Assuming each attention module is followed by $\ell_{\text{FFN}}$ layers of FFN (typically $\ell_{\text{FFN}}$ is small), the overall time and space complexity of a single forward propagation of the student model are $O(\ell \times n^2d + \ell \times \ell_{\text{FFN}} \times nd^2) = O(\ell \times n^2)$ and $O(\ell \times n^2 + \ell \times nd + \ell \times \ell_{\text{FFN}} \times nd) = O(\ell \times n^2)$, respectively.

## E    Experimental Details

### E.1    Dataset Statistics

In Table 5, we show the detailed statistics of seven datasets.

Table 5: The statistics of the datasets.

| Dataset | #Nodes | #Edges | #Features | #Classes | Hom.ratio | Train/Val/Test |
|---------|--------|--------|-----------|----------|-----------|----------------|
| Cora | 2708 | 5429 | 1433 | 7 | 0.81 | 140/500/1000 |
| Citeseer | 3327 | 4732 | 3703 | 6 | 0.74 | 120/500/1000 |
| Pubmed | 19717 | 44338 | 500 | 3 | 0.8 | 60/500/1000 |
| Actor | 7600 | 33544 | 931 | 5 | 0.22 | 3648/2432/1520 |
| Cornell | 183 | 295 | 1703 | 5 | 0.30 | 87/59/37 |
| Texas | 183 | 309 | 1703 | 5 | 0.11 | 87/59/37 |
| Wisconsin | 251 | 499 | 1703 | 5 | 0.21 | 120/50/81 |

### E.2 Implementation Details

In our experiments, we use cross-entropy as the distance metric. For the LGT model, we adopt the GT model (Dwivedi & Bresson, 2020), which computes attention by considering only the first-order neighborhood (including the node itself). As for the GGT model, we employ the vanilla Transformer model(Vaswani et al., 2017). Specifically, the implementation of the Transformer layer in the GGT model is identical to that of the encoder's Transformer layer in the original Transformer.

We maintain fixed values for certain hyperparameters: the pre-training epochs of the teacher model are set to 200, the training epochs of the student model to 500, and the weight decay to 5e-4. We conduct hyperparameter tuning for other important hyperparameters on each dataset using grid search. The hyperparameter ranges are presented in Table 6. We provide the specific configurations of hyperparameters on each dataset in Table 7. We implement our method with Python and PyTorch and use Adam as the optimizer. All experiments are conducted on one GeForce RTX 4090 GPU.

Table 6: The searching space of hyperparameters

| Parameter Names | Hyperparameter Ranges |
| --- | --- |
| hidden size | 64, 128, 256 |
| head number | 1, 2, 4 |
| dropout ratio | 0.3, 0.4, 0.5, 0.6 |
| $k$ | 10, 15, 20, 25, 30, 35, 40, 45, 50 |
| $ep^{(begin)}$ | 0, 50, 100, 150, 200, 250, 300 |
| $ep^{(end)}$ | 100, 200, 300, 400, 500 |
| $\gamma^{(begin)}$ | 2, 3, 4, 5, 6, 7, 8 |
| $\gamma^{(end)}$ | 1, 2, 3, 4 |
| $\beta^{(begin)}$ | 0, 0.1, 0.2, 0.3, 0.4, 0.5, 0.6, 0.7, 0.8, 0.9, 1 |
| $\beta^{(end)}$ | 0.2, 0.4, 0.6, 0.8, 1, 1.5, 2 |
| learning rate | 0.01,0.005,0.001 |

## F Additional Results and Analysis

### F.1 Comparative Analysis of Our Method Against Other Distillation Methods

In this subsection, we conduct a comprehensive comparison between our method and traditional knowledge distillation methods. Specifically, we select two well-established methods, KD (Hinton et al., 2015) and FitNets (Romero et al., 2014), and perform a detailed evaluation against our method.

To ensure a fair and meaningful comparison, we consistently employ LGT as the teacher model, ensuring their performances are comparable. As for the student model, we employ GGT throughout the experiments. The results are shown in Table 8.

The results lead to a clear conclusion: our method significantly outperforms KD and FitNets in terms of performance. On average, across all datasets, our method demonstrates an absolute performance improvement of 6.39% and 7.14% compared to KD and FitNets, respectively. More specifically, on the Cora dataset, our method achieves an absolute performance improvement of 7.36% and 7.51% over KD and FitNets, respectively, further confirming the outstanding performance of our method in practical tasks.

### F.2 Hyperparameter Studies

In this subsection, we investigate the impact of hyperparameters $\gamma$ and $\beta$ and their trade-off with the supervised loss on the student model's performance across the Cora and Citeseer datasets. In total,

Table 7: Optimal hyperparameter configurations across seven graph datasets

| Datasets | Type | Hyperparameters |
|---|---|---|
| Cora | Teacher | hidden size = 128, layers = 2, head number = 2, dropout = 0.5, lr = 0.001, weight decay = 0.0005, |
| | Student | hidden size = 128, layers = 2, head number = 2, dropout = 0.6, lr = 0.005, weight decay = 0.0005, $k = 25$, $\gamma^{((begin)} = 3$, $\gamma^{(end)} = 1.5$, $\beta^{(begin)} = 0.1$, $\beta^{(end)} = 0.4$, ep$^{(begin)} = 0$, ep$^{(end)} = 400$ |
| Citeseer | Teacher | hidden size = 128, layers = 2, head number = 2, dropout = 0.4, lr = 0.003, weight decay = 0.0005, |
| | Student | hidden size = 128, layers = 2, head number = 2, dropout = 0.7, lr = 0.01, weight decay = 0.0005, $k = 15$, $\gamma^{((begin)} = 8$, $\gamma^{(end)} = 2.5$, $\beta^{(begin)} = 0$, $\beta^{(end)} = 1.5$, ep$^{(begin)} = 50$, ep$^{(end)} = 150$ |
| Pubmed | Teacher | hidden size = 128, layers = 2, head number = 1, dropout = 0.3, lr = 0.001, weight decay = 0.0005, |
| | Student | hidden size = 256, layers = 2, head number = 1, dropout = 0.3, lr = 0.005, weight decay = 0.0005, $k = 20$, $\gamma^{((begin)} = 2.5$, $\gamma^{(end)} = 0.5$, $\beta^{(begin)} = 0$, $\beta^{(end)} = 0.2$, ep$^{(begin)} = 0$, ep$^{(end)} = 50$ |
| Actor | Teacher | hidden size = 128, layers = 2, head number = 2, dropout = 0.5, lr = 0.001, weight decay = 0.0005, |
| | Student | hidden size = 256, layers = 2, head number = 2, dropout = 0.6, lr = 0.005, weight decay = 0.0005, $k = 30$, $\gamma^{((begin)} = 5$, $\gamma^{(end)} = 3.5$, $\beta^{(begin)} = 0.2$, $\beta^{(end)} = 2.5$, ep$^{(begin)} = 0$, ep$^{(end)} = 50$ |
| Cornell | Teacher | hidden size = 128, layers = 2, head number = 2, dropout = 0.5, lr = 0.001, weight decay = 0.0005, |
| | Student | hidden size = 256, layers = 2, head number = 2, dropout = 0.5, lr = 0.01, weight decay = 0.0005, $k = 10$, $\gamma^{((begin)} = 1.5$, $\gamma^{(end)} = 1.5$, $\beta^{(begin)} = 0.2$, $\beta^{(end)} = 0.8$, ep$^{(begin)} = 300$, ep$^{(end)} = 500$ |
| Texas | Teacher | hidden size = 128, layers = 2, head number = 2, dropout = 0.5, lr = 0.005, weight decay = 0.0005, |
| | Student | hidden size = 256, layers = 2, head number = 2, dropout = 0.6, lr = 0.01, weight decay = 0.0005, $k = 10$, $\gamma^{((begin)} = 6$, $\gamma^{(end)} = 2.5$, $\beta^{(begin)} = 1$, $\beta^{(end)} = 2.5$, ep$^{(begin)} = 100$, ep$^{(end)} = 400$ |
| Wisconsin | Teacher | hidden size = 128, layers = 2, head number = 2, dropout = 0.3, lr = 0.001, weight decay = 0.0005, |
| | Student | hidden size = 128, layers = 2, head number = 2, dropout = 0.5, lr = 0.01, weight decay = 0.0005, $k = 10$, $\gamma^{((begin)} = 1.5$, $\gamma^{(end)} = 0.5$, $\beta^{(begin)} = 0.3$, $\beta^{(end)} = 0.5$, ep$^{(begin)} = 50$, ep$^{(end)} = 100$ |

Table 8: Performance comparison across knowledge distillation methods. T: Teacher model, S: Student model, Distilled: Distilled model.

| type | KD | | | FitNets | | | Ours | | |
|---|---|---|---|---|---|---|---|---|---|
| | T | S | Distilled | T | S | Distilled | T | S | Distilled |
| Cora | 77.40 | 55.90 | 77.34 | 77.40 | 54.80 | 77.19 | 77.40 | 54.24 | **84.70** |
| Citeseer | 68.60 | 55.80 | 70.27 | 68.6 | 57.5 | 69.51 | 68.05 | 50.60 | **76.40** |
| Pubmed | 77.60 | 73.30 | 78.12 | 77.8 | 77.2 | 78.42 | 79.05 | 72.50 | **83.10** |
| Actor | 37.17 | 37.50 | 37.78 | 37.17 | 37.04 | 38.20 | 36.87 | 35.97 | **39.40** |
| Cornell | 70.27 | 59.46 | 63.78 | 70.27 | 67.57 | 62.16 | 68.46 | 65.76 | **77.03** |
| Texas | 78.38 | 75.68 | 75.14 | 78.38 | 75.68 | 77.57 | 76.57 | 77.03 | **85.58** |
| Wisconsin | 76.47 | 80.39 | 80.98 | 76.47 | 78.43 | 75.10 | 75.48 | 77.55 | **81.95** |

we conduct three sets of studies: a) We vary the values of $\beta$ while keeping $\gamma$ fixed; b) We vary the values of $\gamma$ while keeping $\beta$ fixed; c) We investigate the impact of $r = \frac{\gamma}{\beta} > 0$ by fixing the ratio $r$ and adjusting $\beta$ accordingly.

Figure 4 illustrates the variation in the student model's performance concerning the value of $\beta$ while keeping $\gamma$ fixed. From the results, we can observe that: a) The curves generally exhibit a gradually decreasing trend or a slight increase followed by a decrease. This indicates that for a fixed $\gamma$ value, there exists an appropriate threshold for $\beta$. The student's out-of-class exploration yields benefit only when $\beta$ does not exceed this threshold. If the out-of-class exploration exceeds the range supported by the in-class knowledge, the performance of the student model will decline; b) If $\gamma$ slowly increases, the downward trend of the curves is alleviated, suggesting that improving in-class guidance facilitates the student's ability to engage in more extensive out-of-class exploration.

Figure 5 illustrates the variation in the performance of the student model concerning the value of $\gamma$ while keeping $\beta$ fixed. It can be seen that increasing $\gamma$ while keeping $\beta$ constant contributes to improving performance. As the teacher's in-class guidance exceeds a threshold, out-of-class exploration becomes increasingly effective. However, an excessively high proportion of in-class guidance may lead to performance degradation, as out-of-class exploration could be marginalized. Furthermore, comparing the curves shows that higher values of $\beta$ may require a correspondingly higher $\gamma$, indicating a more substantial level of teacher guidance to elevate the student's performance from a lower level.

The study of the relationship between $r = \frac{\gamma}{\beta} > 0$ and performance essentially explores the balance between supervised loss and distillation loss, investigating the benefits of applying distillation over naive supervised learning. From Figure 6, it can be observed that regardless of the value of $r$, when $\beta$ is set to 0, $\gamma$ must also be 0. In this case, the student model relies solely on the supervised loss for learning, resulting in suboptimal performance. When $r$ is small, the student receives weak in-class guidance during out-of-class exploration, so applying distillation on top of supervised learning does not improve performance, and higher values of $\beta$ lead to worse performance. As $r$ is appropriately increased, with an increasing $\beta$, the curves undergo a transition from a decreasing trend to an increasing-then-decreasing trend. The initial performance improvement indicates that distillation becomes effective with enhanced in-class guidance. However, the subsequent performance decline is because the student primarily focuses on out-of-class exploration, and distillation cannot yet replace the supervised loss. As $r$ further increases, the decreasing trend of the curves gradually slows down, indicating a weakening effect of supervised loss. When $r$ is relatively large, the downward trend of the curves stabilizes, indicating that the teacher's in-class guidance can somewhat substitute for supervised loss, enabling the student to learn in an unsupervised manner.

## F.3 INTERPRETABILITY STUDIES

To investigate the impact of our method on enhancing the global homophily of the GGT student model during training, we plot the curves showing the average node homophily variation over time (epochs) and compare them with those of the unguided GGT student model (Figure 7).

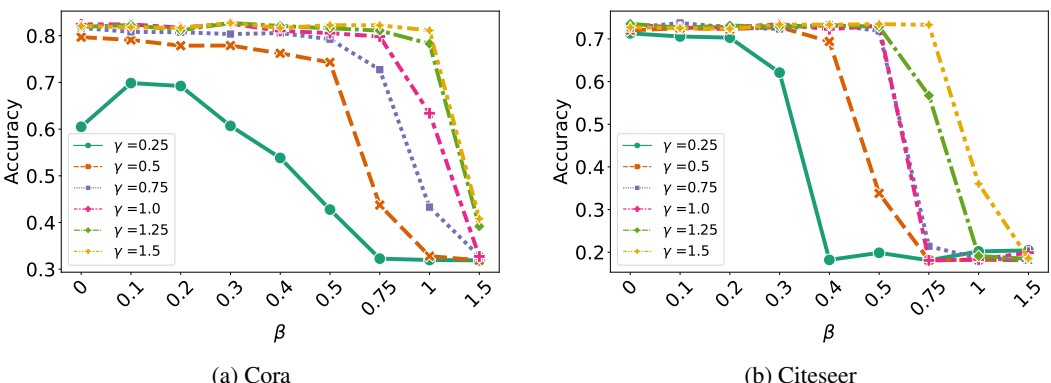

(a) Cora  (b) Citeseer

Figure 4: The impact of hyperparameter $\beta$ on the performance of the student model.

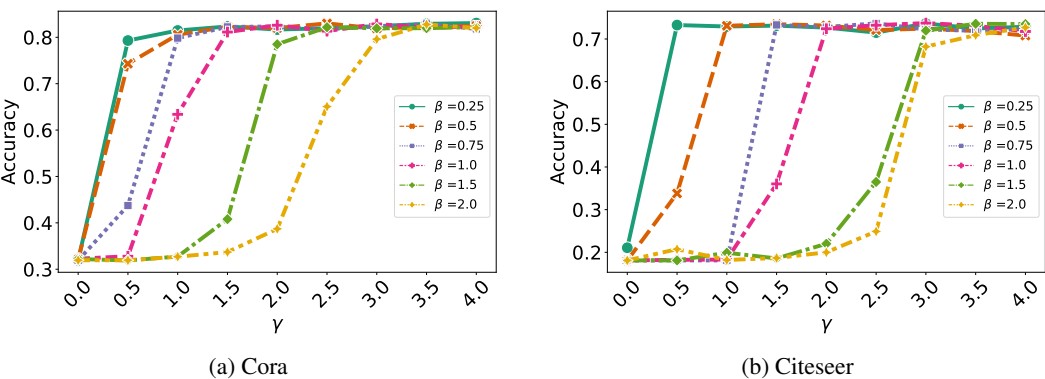

(a) Cora  (b) Citeseer

Figure 5: The impact of hyperparameter $\gamma$ on the performance of the student model.

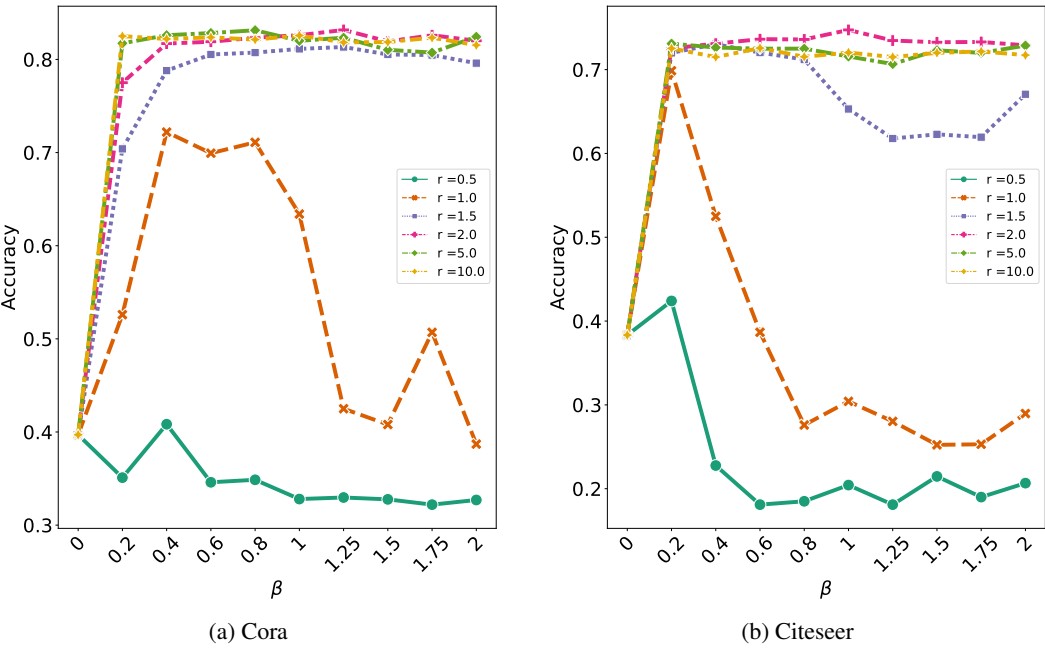

(a) Cora  (b) Citeseer

Figure 6: The impact of hyperparameter ratio $r$ on the performance of the student model.

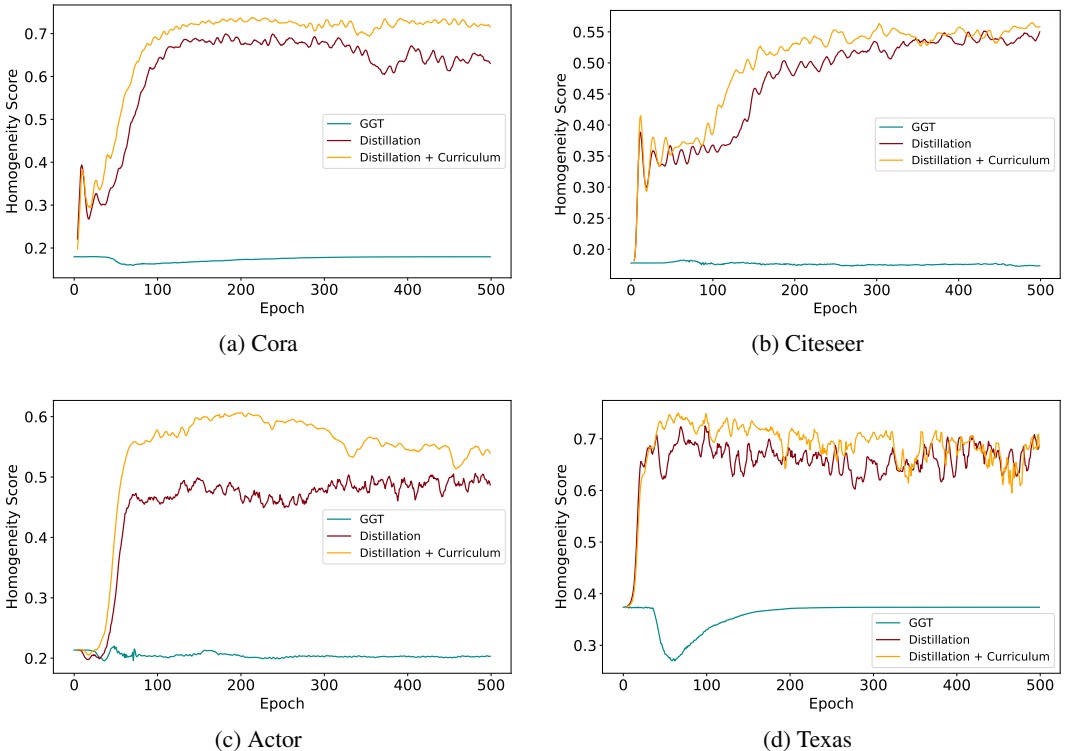

(a) Cora

(b) Citeseer

(c) Actor

(d) Texas

Figure 7: Evolution of average global homophily for GGT student models: a comparison of unguided, guided, and curriculum-enhanced.

From Figure 7, it can be observed that the unguided student struggles to discover beneficial interactions globally, leading to limited realization of its potential and overfitting. However, with teacher guidance, the student's capacity for independent exploration significantly improves, resulting in a notable performance enhancement. Furthermore, incorporating curriculum design further enhances the student's learning outcomes.

### F.4 VISUALIZATION

In this subsection, we present scatter plots of the original features and embeddings produced by GCN and our method on the Cora and Citeseer datasets (Figure 8). The scatter plots show the following: a) The original node features appear messy and exhibit cross-class fusion. b) The embeddings generated by our method demonstrate more explicit classification boundaries and better separability than those produced by GCN.

## G FURTHER DISCUSSIONS

In this paper, we propose attention distillation for Graph Transformers, presenting a novel method to the field of graphs. While similar works exist in NLP, such as MINILM (Wang et al., 2020), which proposes distilling the attention coefficients of the last Transformer layer of the teacher model to the student model, and MobileBERT (Sun et al., 2020), which performs attention distillation layer-to-layer, their motivations differ significantly from ours. We primarily focus on GT models, aiming to improve their training, mitigate overfitting, and enhance generalization abilities based on their unique characteristics. We address a specific and specialized problem, whereas previous research mainly focuses on model compression for generic models.

Further, we introduce "curriculum distillation" in our work. Although the term has been used in previous works, our method differs significantly in context, motivation, and methodology from those earlier studies. For example, this paper (Li et al., 2023) employs temperature control in the Softmax

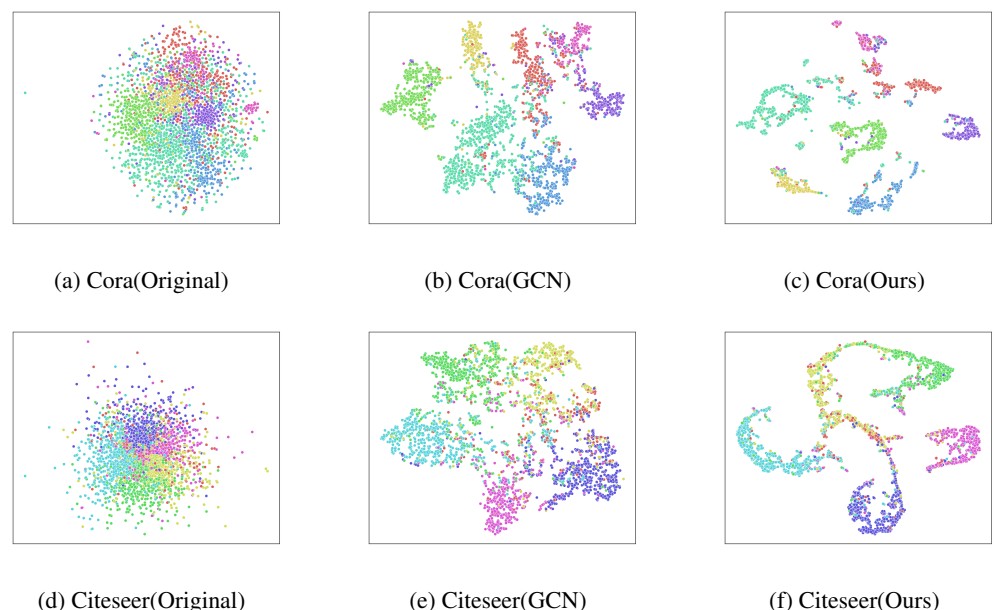

(a) Cora(Original)        (b) Cora(GCN)        (c) Cora(Ours)

(d) Citeseer(Original)        (e) Citeseer(GCN)        (f) Citeseer(Ours)

Figure 8: Scatter plots of original node features and embedding produced by GCN and our method on the Cora and Citeseer datasets.

operator of the teacher's output to gradually transit from fuzzy to clear teacher guidance, focusing on model compression. It provides a relatively simple student model with knowledge ranging from easy to complex. In contrast, we tackle the challenge of overfitting in a complex student model. Our curriculum design captures a dynamic balance between strict guidance and independent innovation over time. In terms of methodology, previous works tend to "constrain" the teacher initially and gradually "release" it, while our method adopts a different method by "strictly guiding" the student initially and allowing for "innovation" later. Therefore, these two methods differ significantly in context, motivation, and methodology.

## H    THE EXTENSION OF OUR METHOD TO GNNS

While initially designed for Transformers and not specifically tailored for GNNs, our method is readily adaptable and extendable to various GNN models, including GCN (Kipf & Welling, 2017) and SGC (Wu et al., 2019), with minor adjustments. Specifically, we can generate similarity matrices for non-attention models using their node embeddings. These matrices serve as rough approximations of the attention matrices within our knowledge distillation framework.

To illustrate this adaptability, we present a preliminary experiment distilling knowledge from GCN to GGT. The corresponding experimental results are summarized in Table 9.

Table 9: Distillation performance comparison.

| Model | Cora | Citeseer | Pubmed | Actor | Cornell | Texas | Wisconsin |
|---|---|---|---|---|---|---|---|
| Teacher (GCN) | $82.06 \pm 0.69$ | $71.60 \pm 0.32$ | $79.58 \pm 0.26$ | $27.88 \pm 0.07$ | $53.51 \pm 1.21$ | $69.19 \pm 1.21$ | $57.25 \pm 1.25$ |
| Student (GGT) | $54.24 \pm 3.16$ | $50.60 \pm 3.83$ | $72.50 \pm 0.87$ | $35.97 \pm 0.29$ | $65.76 \pm 5.62$ | $77.03 \pm 2.70$ | $77.55 \pm 3.93$ |
| GCN $\rightarrow$ GGT | $\mathbf{82.90 \pm 0.76}$ | $\mathbf{72.46 \pm 0.95}$ | $\mathbf{81.73 \pm 0.92}$ | $\mathbf{38.08 \pm 0.60}$ | $\mathbf{72.97 \pm 2.20}$ | $\mathbf{81.08 \pm 2.20}$ | $\mathbf{79.90 \pm 1.87}$ |

The experimental results indicate that applying our method to knowledge distillation from GCN to GGT can significantly improve the performance of the student model across all datasets, with an average absolute performance gain of $10.78\%$. Specifically, on the Cora dataset, the absolute performance gain reaches $28.66\%$ compared to the unguided student model, demonstrating the superiority of our method.

However, it can be observed that the performance of "GCN → GGT" still exhibits a certain performance gap compared to the "LGT → GGT" employed in the main text (refer to Table 3 for performance details). We attribute this performance gap to the inherent advantages of using Transformer models as both teacher and student models. In transformer models, the internal attention coefficients are automatically and inherently learned, eliminating the need to generate additional relational structures based on embeddings, as required in GCN. This ensures the reliability of attention mechanisms, as the teacher model strictly follows these learned coefficients for aggregation. In contrast, considering other models (such as GCN), the similarity matrix is derived from their node embeddings in a post-hoc manner, and thus, its reliability, interpretability, and quality heavily depend on and are restricted by the embeddings. Therefore, subsequent construction of relational structures (e.g., similarity matrix) cannot be guaranteed to be equally reliable, interpretable, and high-quality. From this perspective, inherent attention-based models might be more suitable and can benefit more from our distillation framework.

