# OpenReview forum: "Training Graph Transformers via Curriculum-Enhanced Attention Distillation"
_ICLR.cc/2024/Conference — ICLR 2024 poster_

### Official Review · Reviewer_imQL · 2023-10-30

**Soundness:** 2 fair
**Presentation:** 3 good
**Contribution:** 3 good
**Rating:** 5
**Confidence:** 5

**Summary:**

The paper addresses the challenge of training Graph Transformers for semi-supervised node classification with limited labeled data. The authors propose a curriculum-enhanced attention distillation method that utilizes a Local GT teacher and a Global GT student. By introducing the concepts of in-class and out-of-class, and incorporating out-of-class entropy and top-k pruning, the authors facilitate the student's exploration under the teacher's guidance. Inspired by human learning, the method gradually allows for more out-of-class exploration while maintaining a dynamic balance. Extensive experiments demonstrate the effectiveness of the approach, surpassing state-of-the-art methods on seven public graph benchmarks.

**Strengths:**

1. The authors identified the limitations of GTs in the task of semi-supervised node classification and effectively addressed them through knowledge distillation, which is a reasonable and effective approach.
2. The proposed two significant improvements, out-of-class entropy and top-k pruning, can constrain the student’s behavior during out-of-class exploration and enhance its generalization capability.
3. The experiments demonstrate enhanced performance and improved generalization capability of the method.

**Weaknesses:**

1. My major concern is the motivation of the distillation of GTs. Why not apply distillation on other graph networks to obtain more significant performance, e.g. GCNs? Can the proposed method be applied on other networks?
2. In Ablation, which components are the most important? Why "w.o. $\gamma$" , "Global uniform distribution" bring such damage for the performance?
3. Why not compare to some general KD methods for classification task, e.g. KD, AT, FitNets, DKD, CRD?
4. Please provide the details of Teacher and student, including architecture, parameters, etc. If the utilized "Graph Transformer" equal to "Graphormer" or "Nodeformer"? If not, why not conduct distillation on them?

**Questions:**

Please refer to Weaknesses.

---

> ### Author Response · Authors · 2023-11-23
> **Response to reviewer imQL (1/3)**
>
> Thanks for your valuable comments. Below please find our explanation to the listed  weaknesses.
> > **Q1**:My major concern is the motivation of the distillation of GTs. Why not apply distillation on other graph networks to obtain more significant performance, e.g. GCNs? Can the proposed method be applied on other networks?
>
> **A1**:Thank you for the question.
>
> First of all, our proposed method can surely be applied on other graph networks, including GCNs.
>
>
> The motivation behind distilling GTs lies in our aim to enhance the training of Graph Transformer models through knowledge distillation, with the ultimate goal of unleashing the potential of Transformer models in node classification tasks, surpassing the performance of GNN models. Therefore, we choose Transformer models as both teacher and student models, aligning strictly with our motivation. In other words, we do not prefer GNNs to ensure a fair comparison. We aim to show the outstanding potential of Transformer-based or attention-based models enhanced by our proposed distillation framework.
>
>
> While initially tailored for Transformers and not specifically intended for GNNs, our method is easily adaptable and extendable to other GNN models, including GCN [1] and SGC [2], with some necessary minor modifications. Specifically, we can generate similarity matrices for non-attention models via their node embeddings. These matrices can be rough approximations of the attention matrices in our knowledge distillation framework.
>
>
> Below are the results of our preliminary experiments involving distillation from GCN to GGT, GCN to GCN, LGT to GCN, LGT to GGT.
>
> | **Method** | **Cora** | **Citersser** |
> |:--------:|:----------:|:----------:|
> | Vanilla GCN     | 82.06 $\pm$ 0.69 | 71.6 $\pm$ 0.32 |
> | Vanilla LGT      | 77.40 $\pm$ 0.56 | 68.05 $\pm$ 1.06 |
> | Vanilla GGT      | 54.24 $\pm$ 3.16 | 50.60 $\pm$ 3.83 |
> | GCN $\rightarrow$ GGT      | 82.90 $\pm$ 0.76 | 72.46 $\pm$ 0.95 |
> | GCN $\rightarrow$ GCN      | 82.28 $\pm$ 0.37 | 72.43 $\pm$ 0.35  |
> | LGT $\rightarrow$ GCN      | 82.85 $\pm$ 0.85 | 73.73 $\pm$ 0.55 |
> | LGT $\rightarrow$ GGT      | **84.70 $\pm$ 0.56** | **76.40 $\pm$ 0.70** |
>
> The above results indicate that all four distillation methods can lead to a certain degree of performance improvement in the student model. However, our method achieved the best results on both datasets, significantly outperforming other variants. We attribute this performance gap to the inherent advantages of using Transformer models as both teacher and student models.
>
> As mentioned and discussed in Section 3.1 of the main text, the internal attention coefficients in Transformer models are automatically and inherently learned, eliminating the need to generate additional relational structures based on embeddings as required in GCNs. This ensures the reliability of attention mechanisms, as the teacher model strictly follows these learned coefficients for aggregation. In contrast, considering other models (such as GCN), the similarity matrix is derived from their node embeddings in a post-hoc manner, and thus, its reliability, interpretability, and quality heavily depend on and are restricted by the embeddings. That's why subsequent construction of relational structures (e.g., similarity matrix) cannot be guaranteed to be equally reliable, interpretable, and high-quality. From this perspective, inherent attention-based models might be more suitable and can benefit more from our distillation framework, just as consistent larger performance gains shown in the above table (average absolutely 2.35\% improvement for the listed datasets).
>
> We sincerely thank you for your advice again, and we have included all the discussion in Appendix H of this revision.

---

> > ### Author Response · Authors · 2023-11-23
> > **Response to reviewer imQL (2/3)**
> >
> > > **Q2**: In Ablation, which components are the most important? Why "w.o. $\gamma$", "Global uniform distribution" bring such damage for the performance?
> >
> > **A2**: In ablation, attention distillation emerges as the most crucial component in our method. The significance of attention distillation is thoroughly analyzed in Section 3.1. Also we list the reason as below:
> >
> > Attention distillation, characterized by the transfer of attention coefficients from the teacher to the student model, serves as a foundational component in our method. It enables the student to leverage the teacher’s acquired attention patterns. This critical process allows the student model to inherit and benefit from the teacher's learned attention patterns, enabling the student to utilize the acquired implicit knowledge in out-of-class exploration to discover more valuable interactions. Consequently, it plays a central role in enhancing the student's ability to capture important features in the data.
> >
> >
> > The ablation experiment "w.o. $\gamma$" involves setting the hyperparameter $\gamma$ to 0. In this case, the loss function comprises only the supervised loss and out-of-distribution entropy, essentially eliminating the distillation loss. This implies that the student receives no guidance from the teacher. Driven by out-of-distribution entropy, the student engages in out-of-class exploration without acquiring much beneficial knowledge. This impedes model training and results in a performance decline. This ablation experiment emphasizes the importance of teacher guidance and demonstrates that the utilization of out-of-distribution entropy is contingent on teacher guidance.
> >
> > As for the "Global uniform distribution”, this ablation involves replacing all the teacher model's attention coefficients with a global uniform distribution, eliminating the effective guidance provided by the teacher model to the student model. By only providing a uniform distribution, the student struggles to learn implicit knowledge from valuable neighbor interactions, hindering its ability to discover distant yet valuable interactions during out-of-class exploration. Therefore, guidance that lacks valuable knowledge and is almost equivalent to having no guidance (removing the teacher model and letting the student train independently) does not lead to significant improvements and may even be detrimental. This ablation experiment helps us realize this point, confirming that the crucial factor for performance improvement lies in the guidance provided by the teacher.
> >
> > > **Q3**: Why not compare to some general KD methods for classification task, e.g. KD, AT, FitNets, DKD, CRD?
> >
> > **A3**: We appreciate the reviewer's insightful question.
> >
> > Indeed, we have conducted a comparison with several general knowledge distillation methods, including KD and FitNets. The experimental results on the Cora and Citeseer datasets are listed as below:
> >
> > | Datasets / Models | Teacher  |  Student |  Distilled |
> > |-------------------|----------|----------|------------|
> > | **Cora**          |          |          |            |
> > | KD                |   77.40  |   55.90  |   77.34    |
> > | FitNets           |   77.40  |   54.80  |   77.19    |
> > | Ours              |   77.40  |   54.24  | **84.70** |
> > | **Citeseer**      |          |          |            |
> > | KD                |   68.60  |   55.80  |   70.27    |
> > | FitNets           |   68.60  |   57.50  |   69.51    |
> > | Ours              |   68.05  |   50.60   |  **76.40** |
> >
> > From the experimental results, it can be observed that on the Cora and Citeseer datasets, KD achieves an average absolute performance improvement of 17.95\% compared to the student model without guidance. Similarly, FitNets also achieves an average absolute performance improvement of 17.2\%. In general, distilling knowledge from a teacher model by using its outputs or intermediate layer representations to train a student model can facilitate the student's training.
> >
> > However, after careful consideration, we decided not to include these results in the paper at that time. The primary reason for this decision stems from the distinct focus and motivation behind our method, which diverges from the objectives of traditional knowledge distillation methods. Unlike methods primarily designed for model compression, our method specifically addresses challenges related to Graph Transformers, particularly in semi-supervised scenarios.
> >
> > we appreciate the reviewer's suggestion regarding a more comprehensive comparative analysis. In response, we have included the results of some general knowledge distillation methods on multiple datasets and provide a more detailed analysis in Appendix F.1 of the revision. Thank you once again for your suggestion, as it contributes to the improvement of the quality of our paper.

---

> > > ### Author Response · Authors · 2023-11-23
> > > **Response to reviewer imQL (3/3)**
> > >
> > > > **Q4**: Please provide the details of Teacher and student, including architecture, parameters, etc. If the utilized "Graph Transformer" equal to "Graphormer" or "Nodeformer"? If not, why not conduct distillation on them?
> > >
> > > **A4**: For the LGT model, we implement it using a GT [5] model with attention computations limited to the first-order neighborhood of each node, including itself. As for the GGT model, we opt for the vanilla Transformer [6] implementation, where GGT’s Transformer layers are identical to those of the standard Transformer. Both LGT and GGT models have their Transformer layers fixed at 2, and the specific ranges for other hyperparameters can be found in Appendix E.2. In the revision, we have included specific hyper-parameter configurations on each dataset in Appendix E.2.
> > >
> > >
> > > In our model selection, the chosen model is neither Graphormer nor Nodeformer. We opt for simple frameworks to showcase the potential of our method, hence choosing the naive GGT encoders. As long as it aligns with our motivation – using a teacher model with reliable performance to guide a student model with significant potential but susceptible to overfitting – we can replace the student model with other architectures, such as Graphormer [7] and Nodeformer [8]. We attempt to replace the GGT model we use with Graphormer and SAT [9] (another type of Global Graph Transformer), and the experimental results are presented below. Due to Nodeformer not explicitly constructing attention matrices, implementation requires more advanced techniques, which we plan to explore in the future. It can be observed that, compared to unguided student models, our method significantly improves the performance of student models, surpassing that of teachers. This also demonstrates the superiority of our method. We plan to explore the integration of methods and techniques from prior research with our framework in future work, aiming to achieve enhanced performance.
> > >
> > >
> > > | Datasets / Models | Teacher  |  Student |  Distilled |
> > > |-------------------|----------|----------|------------|
> > > | **Cora**          |          |          |            |
> > > | LGT $\rightarrow$ Graphormer   |   78.90  |   56.50  |   **81.18**    |
> > > | LGT $\rightarrow$ SAT          |   78.10  |   60.80  |   79.80    |
> > > | **Citeseer**      |          |          |            |
> > > | LGT $\rightarrow$ Graphormer   |   67.80  |   57.90  |   **73.28**    |
> > > | LGT $\rightarrow$ SAT          |   67.80  |   58.30  |   72.20    |
> > >
> > > Thank you for your valuable suggestion, as it significantly contributes to enhancing the quality of our paper.
> > >
> > > **Reference**
> > >
> > > [1] Thomas N. Kipf and Max Welling. Semi-supervised classification with graph convolutional networks. In International Conference on Learning Representations, 2017.
> > >
> > > [2] Felix Wu, Amauri Souza, Tianyi Zhang, Christopher Fifty, Tao Yu, and Kilian Weinberger. Simplifying graph convolutional networks. In International conference on machine learning, pp. 6861–6871. PMLR, 2019.
> > >
> > > [3] Geoffrey Hinton, Oriol Vinyals, and Jeff Dean. Distilling the knowledge in a neural network. arXiv preprint arXiv:1503.02531, 2015.
> > >
> > > [4] Adriana Romero, Nicolas Ballas, Samira Ebrahimi Kahou, Antoine Chassang, Carlo Gatta, and Yoshua Bengio. Fitnets: Hints for thin deep nets. arXiv preprint arXiv:1412.6550, 2014.
> > >
> > > [5] Vijay Prakash Dwivedi and Xavier Bresson. A generalization of transformer networks to graphs. arXiv preprint arXiv:2012.09699, 2020.
> > >
> > > [6] Ashish Vaswani, Noam Shazeer, Niki Parmar, Jakob Uszkoreit, Llion Jones, Aidan N Gomez, Łukasz Kaiser, and Illia Polosukhin. Attention is all you need. Advances in neural information processing systems, 30, 2017.
> > >
> > > [7] Chengxuan Ying, Tianle Cai, Shengjie Luo, Shuxin Zheng, Guolin Ke, Di He, Yanming Shen, and Tie-Yan Liu. Do transformers really perform badly for graph representation? Advances in Neural Information Processing Systems, 34:28877–28888, 2021.
> > >
> > > [8] Qitian Wu, Wentao Zhao, Zenan Li, David P Wipf, and Junchi Yan. Nodeformer: A scalable graph structure learning transformer for node classification. Advances in Neural Information Processing Systems, 35:27387–27401, 2022.
> > >
> > > [9] Dexiong Chen, Leslie O’Bray, and Karsten Borgwardt. Structure-aware transformer for graph representation learning. In International Conference on Machine Learning, pp. 3469–3489. PMLR, 2022.

---

### Official Review · Reviewer_4ymw · 2023-10-31

**Soundness:** 3 good
**Presentation:** 2 fair
**Contribution:** 3 good
**Rating:** 6
**Confidence:** 4

**Summary:**

The paper introduces a novel curriculum-enhanced attention distillation approach to enhance the training of Graph Transformers (GTs). This method leverages attention coefficients as knowledge representations for more effective knowledge transfer from a limited graph transformer (LGT) teacher to a general graph transformer (GGT) student. The approach also incorporates out-of-class entropy and top-k pruning to promote active exploration and the surpassing of the teacher's performance by the student. Additionally, curriculum distillation is introduced to manage the attention focus dynamically between in-class and out-of-class learning phases, thereby improving the overall performance. The proposed method has been empirically validated across multiple datasets, demonstrating its efficacy in improving the GTs training process and their generalization capabilities

**Strengths:**

- The paper is well-written and presented.
- Empirical results are promising.

**Weaknesses:**

- The novelty is limited. Not a big concern, but the paper combines a few well-known methods.
- The approach is a bit complicated.

**Questions:**

- The loss function is complex with a few components. In Table 4 the authors evaluated their pipeline's performance sensitivity to each of the components. My first question is how sensitive the model is to variation of these hyperparameters (e.g. $\gamma$, $\beta$)? Both in case of stability and performance.
- Can authors please comment on the scalability of this model? Does it scale to bigger datasets? If yes, have they tried running their model on those?
- Have the authors witnessed a difference in student or teacher behavior in datasets with relatively high homophily values differences?

---

> ### Author Response · Authors · 2023-11-23
> **Response to reviewer 4ymw**
>
> We thank the reviewer for the insightful comments. We reply to all your comments below.
> > **Q1**:The loss function is complex with a few components. In Table 4 the authors evaluated their pipeline's performance sensitivity to each of the components. My first question is how sensitive the model is to variation of these hyperparameters (e.g. $\gamma$, $\beta$)? Both in case of stability and performance.
>
>
> **A1**:Thank you for your insightful question. We have conducted a thorough sensitivity analysis of hyperparameters in Appendix F.1 of our submission. In Appendix F.1, we investigate the impact of variations in hyperparameters $\gamma$ and $\beta$ on the model's stability and performance. Specifically, when keeping $\gamma$ fixed at an appropriate value, we observe that increasing $\beta$ leads to a decline in model performance when $\beta$ exceeds a certain threshold. Conversely, when keeping $\beta$ fixed at an appropriate value, we observe that increasing $\gamma$ gradually improves model performance, reaching a plateau after a certain point. Please kindly find a comprehensive analysis in Appendix F.1. We’ll ensure a clear link between Section 4 and Appenxix F.1 in the revision.
>
>
> > **Q2**:Can authors please comment on the scalability of this model? Does it scale to bigger datasets? If yes, have they tried running their model on those?
>
> **A2**:We appreciate the reviewer's insight into the scalability of our method. The scalability of our method depends on the scalability of the chosen student model. We choose the vanilla Transformer [1] as our student model for simplicity. Though the specific chosen student model, i.e. GGT, in our framework has relatively lower scalability, Our method allows for the replacement of the student model effortlessly. For instance, our method can scale to larger graphs by substituting the student model with a more scalable Transformer variant, such as SGFormer [2]. Given the orthogonality and complementarity of our work, we aim to conduct in-depth exploration, investigation, and discussion of these extensions and integrations in future research, accompanied by comprehensive experimental studies. We believe that by incorporating certain methods and techniques from previous research, we can effortlessly enhance scalability and expect further performance improvements.
>
> > **Q3**:Have the authors witnessed a difference in student or teacher behavior in datasets with relatively high homophily values differences?
>
> **A3**:Compared to homogeneous graphs, node classification tasks on heterogeneous graphs have always been considered more challenging, as evident from the performance of traditional GNN models on heterogeneous graphs, as shown in Table 2. However, the framework proposed in this paper demonstrates consistent performance improvements on both homogeneous and heterogeneous graphs. The experimental results in Table 2 indicate that our method consistently outperforms the second-best method on homogeneous and heterogeneous graphs. Upon comparing the performance of teacher and student models, as shown in Table 3, we observe a more significant improvement in heterogeneous graphs. This could be attributed to the synergy between our proposed method and the Transformer model, where the teacher model provides valuable hints to the student model. This allows the student model to explore more beneficial interactions among non-neighbor nodes (i.e., discovering those distant yet potentially beneficial interactions), thus leading to a substantial enhancement in performance.
>
> **Reference**
>
> [1] Ashish Vaswani, Noam Shazeer, Niki Parmar, Jakob Uszkoreit, Llion Jones, Aidan N Gomez, Łukasz Kaiser, and Illia Polosukhin. Attention is all you need. Advances in neural information processing systems, 30, 2017.
>
> [2] Qitian Wu, Wentao Zhao, Chenxiao Yang, Hengrui Zhang, Fan Nie, Haitian Jiang, Yatao Bian, and Junchi Yan. Simplifying and empowering transformers for large-graph representations. arXiv preprint arXiv:2306.10759, 2023.

---

### Official Review · Reviewer_qbbX · 2023-11-04

**Soundness:** 3 good
**Presentation:** 3 good
**Contribution:** 2 fair
**Rating:** 8
**Confidence:** 3

**Summary:**

This paper proposes a novel curriculum-enhanced attention distillation method to improve the training of graph transformers for semi-supervised node classification. The key ideas and contributions are:

- Proposes using attention coefficients from a Local Graph Transformer (LGT) teacher to guide an untrained Global Graph Transformer (GGT) student via layer-to-layer attention distillation. - Introduces the concepts of in-class and out-of-class attention. Proposes two techniques - out-of-class entropy and top-k pruning - to encourage active out-of-class exploration by the student.

- Inspired by curriculum learning, proposes curriculum distillation where the teacher gradually allows more out-of-class exploration via dynamic scheduling of distillation loss weights.

- Achieves significant performance gains over strong baselines on seven benchmark graph datasets, demonstrating the ability to improve training and generalization of graph transformers.

Overall, the work makes multiple innovations in adapting knowledge distillation for graph transformers in a semi-supervised node classification setting. The curriculum-based attention distillation framework is intuitive and achieves strong empirical results.

**Strengths:**

-  The work presents a novel perspective of applying attention-based knowledge distillation for graph transformers. Using the teacher's attention maps to guide the student is an original idea. The curriculum distillation framework that balances in-class and out-of-class attention over time is also a creative approach.

-  The methodology is technically sound, with clear algorithm descriptions, reasonable design choices, and rigorous empirical evaluation. The curriculum scheduling strategies are grounded in educational learning principles. The improvements for out-of-class attention demonstrate thoughtful analysis.

-  The paper is well-written and easy to follow. The motivations are clearly explained, and the methodology sections provide sufficient details. Figures aid understanding of the core concepts like curriculum scheduling.

**Weaknesses:**

- The proposed method relies on selecting the right teacher-student pair, but the criteria for these choices are not fully analyzed. How does performance vary for different teacher-student combinations?

- While outperforming baselines, the absolute performance gaps are sometimes small (~1-2%). The gains may not justify the added complexity in some applications.

- Curriculum scheduling adds hyperparameters like epoch boundaries. The impact of these hyperparameters could be studied more systematically. In addition, I highly recommend that authors use auto-distillation (KD-Zero: Evolving Knowledge Distiller for Any Teacher-Student Pairs (NeurIPS-2023, Automated Knowledge Distillation via Monte Carlo Tree Search (ICCV2023)) to optimize different parameter options.


- Attention distillation for other graph model families besides transformers could also be promising but is not explored. It is essential to incorporate a thorough discussion of relevant KD-related studies, including Self-Regulated Feature Learning via Teacher-free Feature Distillation (ECCV2022), NORM: Knowledge Distillation via N-to-One Representation Matching (ICLR2023), Shadow Knowledge Distillation: Bridging Offline and Online Knowledge Transfer (NIPS2022), DisWOT: Student Architecture Search for Distillation Without Training (CVPR2023) . This discussion will help position the proposed approach within the existing literature, establish connections, and provide valuable insights for potential comparisons.

**Questions:**

see Weaknesses


-------------------------------------------


The author's response addressed my concerns well, so I'm improving my score to acceptance, thanks!

---

> ### Author Response · Authors · 2023-11-23
> **Response to reviewer qbbX (1/2)**
>
> Thank you for your review and suggestions. We reply to all your comments below.
> > **Q1**:The proposed method relies on selecting the right teacher-student pair, but the criteria for these choices are not fully analyzed. How does performance vary for different teacher-student combinations?
>
> **A1**:Thank you for raising a thoughtful question regarding the selection of the teacher-student pair. The motivation behind our work is to leverage a teacher model with decent performance and resistance to overfitting to guide a student model with significant potential but susceptibility to overfitting, enhancing the training of the student model. Given this criterion, there are multiple suitable teacher-student pairs to choose from. In our submission, we propose a generic framework, preferring simple and widely understood architectures to showcase the method's potential. The selected LGT and GGT models are fundamental Transformer encoders, emphasizing that outstanding performance can be achieved with the most basic attention layers. We also have conducted preliminary explorations of different teacher and student model combinations in the ablation studies section (Section 4.4). From the experimental results, it is clear that our method outperforms three variants: "GGT $\rightarrow$ GGT", "Local neighborhood uniform distribution", and "Global uniform distribution", affirming the soundness and rationality of our design.
>
> > **Q2**:While outperforming baselines, the absolute performance gaps are sometimes small (~1-2\%). The gains may not justify the added complexity in some applications.
>
> **A2**:Thank you for the insightful question. While, on certain datasets, the absolute performance improvement of our method over baselines may not be substantial, our method consistently improves performance across all seven datasets, with an average absolute improvement of 4.9\%. Notably, we achieve a significant gain of 7.74\% on the Texas dataset, as detailed in Section 4.2. Different from traditional knowledge distillation methods that mainly focus on model compression, our method presents a novel and interesting motivation, i.e., enabling the student model to surpass the performance of the teacher model (See detailed discussions in Section 4.3 in the main text for a specific distillation comparison), which is definitely ignored by prior works. Additionally, our framework's adaptability and scalability are noteworthy, relying on a proposed loss function that is easy to implement in practice.
>
> > **Q3**:Curriculum scheduling adds hyperparameters like epoch boundaries. The impact of these hyperparameters could be studied more systematically. In addition, I highly recommend that authors use auto-distillation (KD-Zero: Evolving Knowledge Distiller for Any Teacher-Student Pairs (NeurIPS-2023, Automated Knowledge Distillation via Monte Carlo Tree Search (ICCV2023)) to optimize different parameter options.
>
> **A3**:Thank you for the valuable feedback. We find your advice quite helpful and beneficial for our work. While we have discussed some important hyperparameters in Curriculum scheduling in Appendix F.1, we acknowledge that our analysis may not have been exhaustive. We appreciate the suggestion to employ auto-distillation techniques such as KD-Zero [1] and Automated Knowledge Distillation via Monte Carlo Tree Search [2], which could provide a more systematic exploration of hyperparameter options. We will thoroughly explore and incorporate these extensions into our future work. We sincerely appreciate your guidance, and we believe your advice will significantly contribute to refining our research in upcoming studies.

---

> > ### Author Response · Authors · 2023-11-23
> > **Response to reviewer qbbX (2/2)**
> >
> > > **Q4**:Attention distillation for other graph model families besides transformers could also be promising but is not explored. It is essential to incorporate a thorough discussion of relevant KD-related studies, including Self-Regulated Feature Learning via Teacher-free Feature Distillation (ECCV2022), NORM: Knowledge Distillation via N-to-One Representation Matching (ICLR2023), Shadow Knowledge Distillation: Bridging Offline and Online Knowledge Transfer (NIPS2022), DisWOT: Student Architecture Search for Distillation Without Training (CVPR2023). This discussion will help position the proposed approach within the existing literature, establish connections, and provide valuable insights for potential comparisons.
> >
> > **A4**:Thank you for the insightful comment. While we have mentioned some knowledge distillation studies in the graph domain in Appendix A.2, there are indeed more potential avenues to explore. In the revision, we have included the references mentioned by the reviewer and provided a discussion of the differences, similarities, and possible connections between our work and these studies in Appendix A.2. For example, in DisWOT [3], the authors utilize the feature maps of samples to form a correlation matrix and achieve knowledge distillation by minimizing the distance between the correlation matrices of the teacher and student models. This relation distillation technique shares similarities with our method. However, our motivations are not identical. DisWOT aims to improve model compression, whereas we leverage the characteristics of the Graph Transformers to enhance the training process and alleviate overfitting. Once again, we appreciate the valuable advice from the reviewer. We believe your advice will surely enhance the overall quality of our paper.
> >
> > **Reference**
> >
> > [1] Lujun Li, Peijie Dong, Anggeng Li, Zimian Wei, and Yang Ya. Kd-zero: Evolving knowledge distiller for any teacher-student pairs. In Thirty-seventh Conference on Neural Information Processing Systems, 2023a.
> >
> > [2] Lujun Li, Peijie Dong, Zimian Wei, and Ya Yang. Automated knowledge distillation via monte carlo tree search. In Proceedings of the IEEE/CVF International Conference on Computer Vision, pp. 17413–17424, 2023b.
> >
> > [3] Peijie Dong, Lujun Li, and Zimian Wei. Diswot: Student architecture search for distillation without training. In Proceedings of the IEEE/CVF Conference on Computer Vision and Pattern Recognition, pp. 11898–11908, 2023.

---

> ### Comment · Reviewer_qbbX · 2023-11-23
> **Thanks for the response.**
>
> The author's response addressed my concerns well, so I'm improving my score to acceptance, thanks!

---

### Official Review · Reviewer_pnsY · 2023-11-05

**Soundness:** 4 excellent
**Presentation:** 4 excellent
**Contribution:** 4 excellent
**Rating:** 8
**Confidence:** 3

**Summary:**

This paper proposes a curriculum-enhanced attention distillation method that involves utilizing a Local Graph Transformer as the teacher model and a Global Graph Transformer as the student model. They also introduce the concepts of “in-class” and “out-of-class” regions and introduce out-of-class entropy and top-k pruning. Experiments are conducted on 7 node classification benchmarks, which show the performance benefits of the proposed method.

**Strengths:**

The paper proposes an effective method of training Graph Transformers. The method is verified in multiple node classification benchmarks. The distillation comparative studies show that the method generally outperforms both the teacher and student models. The paper is well-written with justified motivations and reasonable solutions.

**Weaknesses:**

Some of the implementation details are unclear. Please see the questions listed below.

**Questions:**

Q1: The distillation losses are calculated by distances between the teacher and student attention coefficients, which are averaged over graph nodes and attention layers, as demonstrated by Equation (7). Does the averaging calculation smooth out the signal?

Q2: What is the exact distance metric $d$ do you use?

Q3: For the top-k pruning, what is the exact $k$ do you use? Does it vary significantly between datasets? For example Pubmed has 19,717 nodes while Cornell and Texas only have 183 nodes, does the k values different?

Q4: What are the specific LGT and GGT models do you use in the experiments?

---

> ### Author Response · Authors · 2023-11-23
> **Response to reviewer pnsY**
>
> We thank the reviewer for the insightful comments and suggestions. Please kindly find our detailed response to each raised question as below.
> > **Q1**:The distillation losses are calculated by distances between the teacher and student attention coefficients, which are averaged over graph nodes and attention layers, as demonstrated by Equation (7). Does the averaging calculation smooth out the signal?
>
> **A1**:Thank you for the in-depth question. Indeed, the averaging calculation is designated to smooth out the signal during knowledge distillation. This process effectively mitigates noise or fluctuations in individual attention coefficients, providing a stable and representative signal for guiding the learning of the student model. Importantly, knowledge distillation is a holistic process, transferring the knowledge from the teacher model to the student model on a global scale. By averaging attention coefficients across all nodes and layers, our method captures essential patterns and relationships globally, significantly enhancing the student model's performance.
>
> > **Q2**:What is the exact distance metric do you use?
>
> **A2**: In our experiments, we use cross-entropy as the distance metric, which is equivalent to the Kullback-Leibler (KL) divergence. We opt for cross-entropy mainly for its simplicity and widespread use in the machine learning community as a common metric for probability distributions. We have emphasized this point in Appendix E.2 of the revision to enhance the paper's clarity.
>
> In addition, most distance metrics are applicable in our experiments, such as Euclidean distance. We notice that different distance metrics may impact our distillation method. An appropriate distance metric has the potential to facilitate better knowledge transfer and improve student training. However, we feel that a more in-depth exploration of this aspect goes beyond the scope of this paper and is orthogonal to our current focus. We plan to explore and extend our framework in future work to delve into this aspect more thoroughly.
>
> > **Q3**:For the top-k pruning, what is the exact $k$ do you use? Does it vary significantly between datasets? For example Pubmed has 19,717 nodes while Cornell and Texas only have 183 nodes, does the $k$ values different?
>
> **A3**:Thank you for raising a crucial question. For top-k pruning, we customize the selection of $k$ values based on dataset size. Smaller datasets like Cornell and Texas utilize $k$ values such as 5,10,15, and 20, while larger datasets have a broader but still constrained range, typically from 10 to 50. Considering that different datasets require different $k$ values, the specific $k$ values are determined through grid search, ensuring the most suitable $k$ for each dataset. For the Pubmed dataset, we set $k=20$, while for the Cornell and Texas datasets, $k=10$. In our submission, we have already included the hyperparameter search space in Appendix E.2. We acknowledge that we overlooked detailing the specific hyperparameter configurations for each dataset in the paper. In the revision, we have provided detailed information regarding the hyperparameter configurations for each dataset in Appendix E.2. We appreciate your guidance and understand the importance of this clarification for reproducibility. These additions will enhance the overall quality of our study. Once again, we express our gratitude for your professional advice.
>
> > **Q4**:What are the specific LGT and GGT models do you use in the experiments?
>
> **A4**:Thank you for addressing a vital question. For the LGT model, we implement it based on the GT model [1], which computes attention by considering only the first-order neighborhood (including the node itself) for each node. As for the GGT model, we adopt the vanilla Transformer architecture [2]. In particular, the implementation of the Transformer layer in our GGT model closely follows that of the vanilla Transformer model. In our model selection, we lean towards straightforward frameworks to demonstrate the potential of our method. The chosen LGT and GGT models are based on the fundamental Transformer encoder, emphasizing the effectiveness achievable with the native attention layer. This choice aligns with our motivation: employing a teacher model with decent performance and stability to guide a student model with significant potential but prone to overfitting. This represents a paradigm distinct from traditional knowledge distillation. We also explore various teacher-student model combinations to validate the rationale and effectiveness of our choices in Section 4.4 of the main text.
>
> **Reference**
>
> [1] Vijay Prakash Dwivedi and Xavier Bresson. A generalization of transformer networks to graphs. arXiv preprint arXiv:2012.09699, 2020.
>
> [2] Ashish Vaswani, Noam Shazeer, Niki Parmar, Jakob Uszkoreit, Llion Jones, Aidan N Gomez, Łukasz Kaiser, and Illia Polosukhin. Attention is all you need. Advances in neural information processing systems, 30, 2017.

---

### Meta-Review · Area_Chair_UagK · 2023-12-17

**Metareview:**

This paper proposes an effective method of training Graph Transformers and verifies it in multiple node classification benchmarks. The distillation comparative studies show that the method generally outperforms both the teacher and student models. Most of the reviewers give positive comments. Reviewer imQL concerns about the distillation of GTs, more ablations, and experiments. The authors give detaileds feedbacks, but the reviewer did not discussed it. The ACs thus decided to accept it.

**Justification For Why Not Higher Score:**

More ablations should be added in the final version.

**Justification For Why Not Lower Score:**

Training graph transformer is worthy to investigated.

---

### Decision · Program_Chairs · 2024-01-16

Accept (poster)